# Large language models predict cognition and education close to or better than genomics or expert assessment
Tobias Wolfram ✉

Previous research using standard social survey data has emphasized a relative lack of power when predicting educational and psychological outcomes. Leveraging a unique longitudinal dataset, we explore predictability of educational attainment, cognitive abilities, and non-cognitive traits. Integrating various measures of computational linguistics and large language model-based embeddings within a SuperLearner framework trained on short aspirational essays written at age 11, we accurately predict cognition and non-cognitive traits at the same and later age to a similar degree as teacher assessments, and better than genomic data. The same is true for predicting final educational attainment. Combining text, genetic markers, and teacher assessments into an ensemble model, we can predict cognitive ability at close to test-retest reliability of gold-standard tests ($R^2_{Holdout} = 0.7$) and explain 38% of individual differences in attainment. A sociological model comparable to the baseline of the Fragile Family Challenge replicates the FFC's findings regarding the level of predictability achievable with such data. These findings show that recent advances in large language models and machine learning equip behavioural scientists with tools for prediction of psycho-social features.

Psychological, life course, and behavioral prediction is an increasingly recognized and timely goal[1–7]. One driving motivation is to identify individuals at risk for negative outcomes and inform the timing and type of intervention, with earlier work focusing on specific outcomes such as the onset of mental health problems[8]. A recent study[9]—the 'Fragile Families Challenge' (FFC)—utilized the 'common task method'[10], which provided important benchmarks on the predictability of life outcomes using a comprehensive set of survey variables and methods: One hundred and sixty research groups leveraged 12,942 social survey variables from the Future of Families and Child Wellbeing Study, making available recent advances in computational power and machine learning algorithms which are increasingly popularized in the social sciences[11]. The most successful predictions explained ~20% of individual differences in Grade Point Average (GPA) and Material Hardship (out-of-sample). For non-cognitive traits such as 'grit', the accuracy was ~5%, showing that a significant portion of variance remained unexplained for the studied outcomes using the predictors at hand.

However, there are methodological developments that promise improvements in predictions. The revolution in natural language processing (NLP) using deep-learning techniques—such as transformer-based language models[12]—has sparked an extraordinary advancement in the way machines can interact with humans and generate human-like language[13].

These developments have not only led to the widespread adoption of NLP in a variety of real-world applications, including 'chatbots', virtual assistants, and automated customer service systems[14], but have also opened up a vast array of new research opportunities. Attention is gradually turning towards more generalizable problems with frontier technologies (e.g., the 'predictability hypotheses' as outlined by ref. [15]), such as analyzing dense data on life events within vector spaces, which represent human life courses. In some domains, epistemic error is being rapidly eliminated in a way which reduces total predictive error to its purely aleatoric component (for a generalized framework, see ref. [16]). It has, for example, been shown that it is feasible to operationalize life courses through a comprehensive registrar of individual, micro-level data which are readable as language and are amenable to NLP-based predictions[17]. Here, we explore the extent to which NLP can help to predict a key part of the educational and psycho-social life course based on alternative, non-standard data sources. We exploit information from aspirational essays written by children at the formative age of eleven, and we compare predictions made from them with genomic techniques and teacher assessments. With this non-standard data, we are able to contrast our predictions with studies such as refs. [9] and [18], which claim that human lives are essentially 'unpredictable'. This is made possible by a birth cohort-based study, which allows such an expansive research design.

Faculty of Sociology, Bielefeld University, Bielefeld, Germany. ✉e-mail: tobias.wolfram@uni-bielefeld.de

Previous text-based attempts have achieved only limited success in predicting, for example, personality, mental health, cognitive ability, and educational achievement, with variance explanations ranging from 5–10%[19–22]. However—in an approach not dissimilar to refs. 23,24 recently demonstrated the potential of essay content and style in predicting authors' outcomes. Using 240,000 admission essays submitted by 60,000 applicants to the University of California, they applied Linguistic Inquiry and Word Count as well as Correlated Topic Modeling to ~1400 words per respondent, resulting in benchmarks of up to 49% variance explained in a composite SAT score and 16% for household income. Such findings suggest that prediction might be possible based on information hidden in alternative data sources in comparison to standard classical survey data.

The sample used in ref. 24 was highly selective and homogenous, comprising only potential college students, which restricts generalizability. SAT scores and household income represent a specific yet limited set of traits that may be highly correlated with textual data. Such findings motivate further research on the potential of text-based prediction, in order to define both the reducible and irreducible 'origins of unpredictability'[18]. We aim to identify how accurate these computationally advanced text-based predictions are for a wider range of psycho-social outcomes—educational attainment and cognitive traits and non-cognitive traits—utilizing recent advances in natural language processing techniques (e.g., ref. 13). We utilize algorithms and computational approaches resulting in combining more than 500 lexicographic metrics used in computational linguistics with high-dimensional, deep-learning-based numerical representations of each text sample. Importantly, our analyses focus on small text samples of around ~250 words only. To maximize predictive accuracy, we use these extracted features as input to an array of established machine learning algorithms within an ensemble SuperLearner framework[25,26]. We repeated algorithmic training and tested in subsamples of our data via the process of cross-validation and reached predictive accuracy on held-out test data while also protecting against overfitting.

We examine predictions across various outcomes and time spans, from contemporaneous outcomes such as cognitive abilities at age 11—when the text samples are collected—to those decades later, such as educational attainment of the children at age 33. We are motivated to study educational attainment as this can be considered the central stratum in the contemporary world[27,28] and is highly consequential for multiple channels of life course trajectories[29–31]. However, the causal effect of educational attainment on individuals' life courses has also been challenged, highlighting the possibility that it is mainly a proxy for effects of educational precursors such as cognitive ability and non-cognitive traits[32]. Therefore, we include these psychological life course predictors in our study. We contrast and combine our text-based prediction with teacher assessments of students and genomic prediction of multiple polygenic scores (see Section Methods). Teacher assessments represent the most common alternative to academic tracking for career prediction[33] and show moderate to high correlations with traits such as academic achievement ($r = 0.65$) and cognitive ability ($r = 0.5$,[34]). These are the current limits to predictive accuracy given our existing understanding of the literature. Investigating the accuracy of prediction with these outcomes is, in general, of great interest to recent meta-theoretical considerations. The comparison and combination of human judgment and machine-based predictions has gained interest not only for macro-economic or political trends, but primarily for individual outcomes such as health diagnostics[35], despite only occasional use in the social sciences[31]. Note that in our comparisons, we also rebuilt the predictive sociological model from the FFC in an analogy between GPA and educational attainment, replicating their results in our data despite the slight measurement difference.

There are also longstanding questions regarding the role of genetics and the (social) environment, which can be answered by considering the interplay of information generated at different parts of an individual's life[36]. Our work speaks to the 'gloomy prospective' of Plomin and others[37] where it is claimed that only genetic variation (and its phenotypical proxies) are useful for prediction[38]. We engage in a direct comparison of genetic prediction—common in science and society[39–41]—with teacher assessment and essay-based predictions. The 'gloomy prospect' of unpredictability

exemplifies a meta-theoretical framework that hypothesizes the complexity-driven unpredictability of historical events, testable through prediction algorithms rather than traditional inferential methods[15,42].

The National Child Development Study [NCDS,[43]] allows us to compare new approaches to prediction in a comparative setting. This British birth cohort data collection—which started in 1958—is still ongoing. It requested participants at age eleven to write an essay of roughly 250 words under the theme of "Imagine you are 25" (see Section Essays). At the same age, teachers of the respondents were asked to assess their students' general knowledge, number work, use of books, and oral ability—all on a scale from one to five—as well as certain behaviors and motor abilities on a scale from one to three (see Section Teacher Evaluations). Eventually, in 2002, blood samples of participants were collected and later genotyped (see Section Genomic Data). Importantly for our research design and the practical goals of behavioral and life course prediction more broadly—e.g., intervention[39]—our measures are time-invariant early-life course predictors, with the genome fixed at conception.

Similar to our text-based analysis, genomics confronts us with the challenge of large, seldom structured data. Genome-wide association studies (GWAS) link molecular genetic variations to psychological and social outcomes[44] and allow out-of-sample prediction from genotyped data based on individual genetic summary scores[45,46]. We follow this approach to reduce the complexity of 35 million genetic markers by applying standard quality control and merging with publicly available summary statistics from highly powered GWAS. We apply a multi-polygenic score (PGS) approach to trait prediction (see Section Methods), leveraging a set of 33 traits which theory would suggest are associated with our outcomes of interest[47]. Our overall research design is schematically displayed in Fig. 1, where we also highlight the temporal element in our predictions.

## Methods
### Data
The dataset used for our analyzes is the National Child Development Study (NCDS), an ongoing British birth cohort study starting with 17,415 children born in a single week of 1958. No new data were collected, and no re-identification or data linkage to individual participants was performed. According to the ethical guidelines of the DGPs/BDP and Bielefeld University's Ethics Review Board, such research does not require formal ethical approval.

We seek to evaluate the extent to which genomic data, teacher assessments, and the information embedded in the essays are able to predict outcome variables that can be broadly categorized under three different themes: cognitive abilities, non-cognitive traits, and socioeconomic outcomes, most of which are directly taken from the NCDS data or created using factor analysis. We briefly describe the construction of all variables involved and refer to the Supplementary Information for a detailed discussion.

**Essays.** At age eleven, study participants were asked to write an essay under the theme "Imagine you are 25". In total 10,511 essays of varying lengths (ranging from 1 to 1239 words) were transcribed. We apply various approaches to extract the maximum amount of information from them. First, by using a pre-trained deep-learning NLP model—`text-embedding-ada-002`—to extract representations of the words of all essays in lower-dimensional space. These embeddings capture the semantic importance of each word in context and allow the expression of the texts by numeric vectors along 1,536 dimensions. In addition, we measure 534 linguistic metrics of lexical diversity [TAALED[48]], lexical sophistication [TAALES[49]] and sentiment [SEANCE[50]], 31 metrics of readability, and grammatical and typographical error/word-ratios. Additional details on the construction of these features are provided in Supplementary Information Section Essays.

**Teacher evaluations.** We use 22 teacher-related variables overall. At age eleven, teachers of the respondents were asked to assess their students' general knowledge, number work, use of books, oral ability on a scale from one to five, as well as certain behaviors and motor abilities on a scale from one to three. In addition, a series of behavioral descriptions was given to a teacher who was

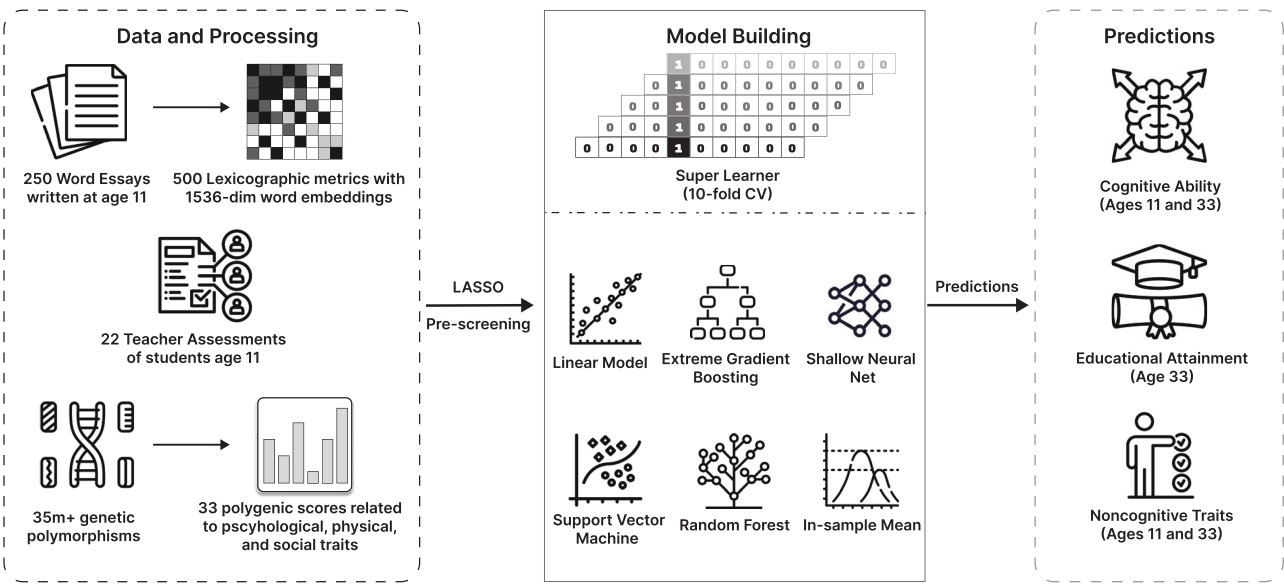

**Fig. 1 | Overview of the research design.** PGSs, essays and teacher assessments are used to construct covariates as input to an ensemble algorithm (SuperLearner) which predicts cognitive ability, non-cognitive traits and educational attainment.

asked to underline the descriptions that best fit the child. By summing the number of selected items, a quantitative assessment of the child's adjustment to school is obtained for multiple categories of behavior. Additional details on the construction of these features are provided in Supplementary Information Section Teacher Assessments.

**Genomic data.** We utilize genotyped data from multiple subsets of NCDS participants. We combine all available genomic data into a single file and restrict the available single-nucleotide polymorphisms (SNPs) to those in common with the reference panel of the 1000 Genomes Project. The final sample contains 37,772,588 variants across 6437 individuals. Using PRSice2[51], we then utilized publicly available summary statistics of genome-wide association studies to construct polygenic scores for a set of 33 curated traits (listed in Table S1). As an exploratory robustness check, the extensive set of polygenic scores provided by the Polygenic Index Repository[52] constructed using LDPred[53] was also tested in preliminary analyzes as a comparison, leading to qualitatively similar results, see Fig. S1. In all but one example (Internalizing Behavior at Age 16), the PGI-Repository scores generated less accurate predictions. These summary statistics span the realms of cognition, mental health, personality, physical composure, social behavior, and substance abuse. Additional details on the construction of these features are provided in Supplementary Information Section Genomic data. All polygenic scores were jointly used as input to the SuperLearner (see below), akin to a multi-polygenic score model[47]. Given our focus on predictive power rather than causal inference, we did not correct for population stratification, as doing so could potentially reduce the predictive utility of the polygenic scores in our specific population. This follows the argument outlined by[54], which considers population stratification as a valid source of genetic variance for polygenic score prediction within a defined population.

**Analytical strategy**
We run the SuperLearner with nested cross-validation, using 5-folds in the inner and 10 in the outer loop. We utilize the Mean Squared Error as the cost function using the `SuperLearner` package in R. No preregistration of our analysis took place.

**Metrics used.** As detailed in our supplementary information, we employed several approaches to extract information from the essays:

- GPT-based embeddings using the `text-embedding-ada-002` model resulting in a 1536-dimensional vector for each essay.
- SALAT-metrics, providing 534 measures related to various linguistic aspects.
- Readability-metrics, consisting of 31 different measures from the `koRpus-package`.
- Grammatical and typographical error ratios, derived using the LanguageTool CLI.

Despite the fact that the main aspects of our analysis were conducted using GPT-3.5, we also compare our results to RoBERTa and GPT 4.0 embeddings. The results—shown in Figure S2—are largely invariant across the GPT-3.5 and 4.0 embeddings, but both offer predictive gains above the RoBERTa embeddings.

**Size of predictive models.** Our main predictive model—as shown in Fig. 2—incorporates all the above different types of text-based features in the SuperLearner framework. The model is large and comprehensive, and as detailed above, uses four key types of extracted information. In Fig. 4, we demonstrate the incremental utility of each individual information set in comparison to other commonly utilized—and simpler—alternatives.

**SuperLearner.** To reduce learning error as much as is reasonably possible with input data from across the three information sources (teacher assessments, essays, genomic data), we utilize a SuperLearner-based approach[25,26]. SuperLearners are ensemble algorithms that estimate the performance of a selection of machine learning models based on cross-validation. The SuperLearner algorithm combines predictions from multiple individual machine learning models, such as Extreme Gradient Boosting, Random Forest, and Support Vector Machines, all in order to generate a final prediction. This is done through a process called 'stacking' or 'model averaging'. We used the default settings provided by the SuperLearner package in R for each algorithm, without performing additional algorithm-specific hyperparameter tuning. This approach focused on the overall ensemble performance while maintaining computational efficiency and reducing overfitting risks. As a large number of variables are extracted from the essays in the form of word embeddings and beyond, LASSO regressions (implemented in `glmnet`) are used as a pre-screening algorithm. LASSO pre-processing was performed once per

**Fig. 2 | 10-fold cross-validated predictive $R^2_{Holdout}$ from SuperLearner models based on essays, teacher assessments and genomic data for various cognitive abilities and non-cognitive traits.** For sample sizes across folds, see Table S5. Whiskers mark lowest and highest $R^2_{Holdout}$ over all CV Folds.

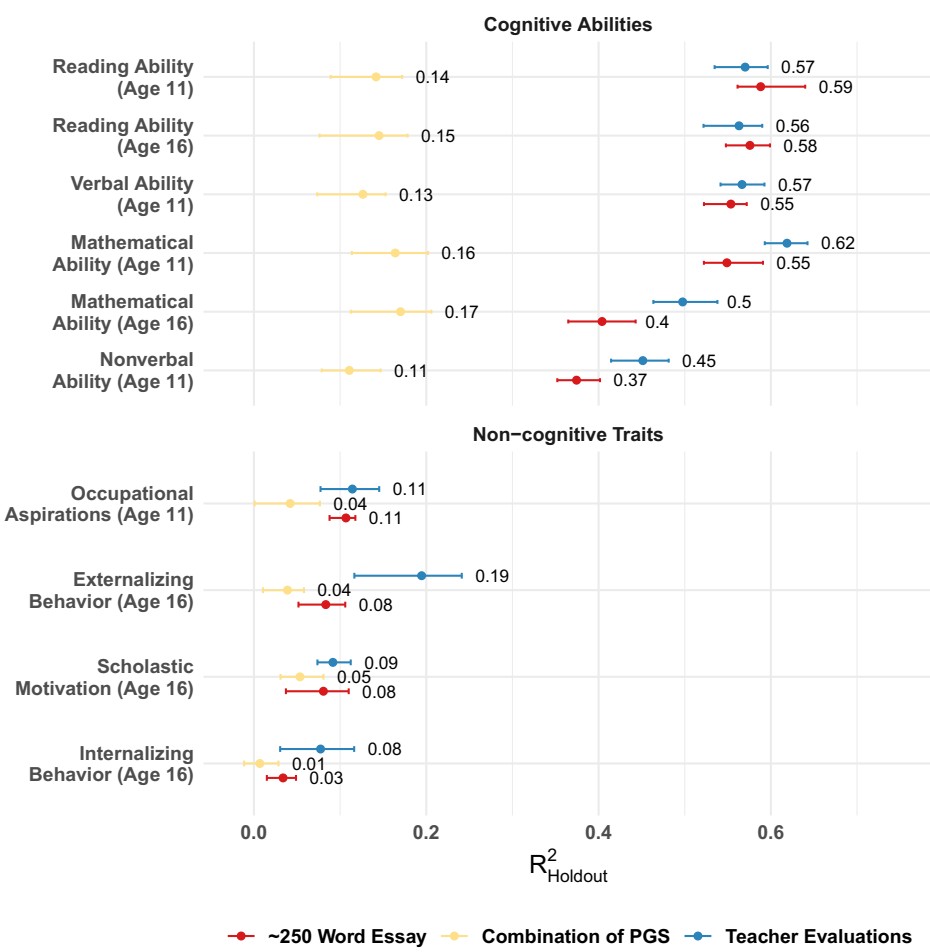

outcome per fold in the cross-validation process. We use a diverse range of machine-learning models as inputs to the SuperLearner: Extreme Gradient Boosting, as implemented in the package `xgboost`[55], RandomForest, as implemented in the package `ranger`[56], a shallow neural network as implemented in the package `nnet`, SVM with a Radial Basis Kernel as implemented in the package `ksvm`, a simple linear regression model in base `R`, and the mean of the outcome in the training set. From these estimates, the SuperLearner determines an optimal weighted average prediction based on the performance of individual composite models. The weights attributed to each of these individual models can be seen in Figure S3, which indicates that it is the Linear Model and the Random Forest, which receive the highest weightings across almost all outcomes.

We used L1 regularization to pre-screen variables in each fold of the SuperLearner. We examined the consistency of feature selection across all folds and outcome variables for each of the three basic SuperLearner models. For the genetic model, all polygenic scores were selected at least once, with household income, educational attainment, and age at first birth being the most consistently selected. In the teacher evaluation model, all covariates were also selected at least once, with general knowledge, number work, and oral ability assessments being selected in every instance. For the essay-based model, while 149 features were never selected, the top 100 most consistently selected features were dominated by embedding dimensions (87) derived from the large language model. Only two non-embedding features from the essays are found among the top 10 most consistently selected: The root type-token ratio and the measure of textual lexical diversity, with the former being selected in 96.6% of all models.

**Cross-validation**. For each outcome variable in each fold, a separate LASSO model was fit to select relevant features. We used 10-fold cross-

validation for all three of our types of models (textual data, genomic data, and teacher assessments). However—as we document in detail in the supplementary material—the performance of a linear model was comparable for genomic data and teacher assessments. The textual data, on the other hand, benefited significantly from the SuperLearner approach, which is why we emphasize the cross-validated results for this model. The number of features retained varied depending on the outcome variable and specific fold.

**Metric assessment**. We use the $R^2_{Holdout}$ metric, also known as the 'predictive $R^2$', as our primary evaluation criterion. This choice aligns with recent related studies in the literature[9]. The $R^2_{Holdout}$ can be considered a predictive counterpart to the well-known coefficient of determination ($R^2$). This metric assesses a model's predictive performance by comparing its predictions on holdout data to a baseline prediction using only the mean of the training set. Specifically, for each cross-validation fold, we calculate:

$$R^2_{Holdout} = 1 - \frac{\sum_{i \in Holdout}(y_i - \hat{y}_i)^2}{\sum_{i \in Holdout}(y_i - \bar{y}_{Training})^2}$$

where $y_i$ is the actual value for observation $i$ in the holdout set $\hat{y}_i$ is the predicted value for observation $i$ $\bar{y}_{Training}$ is the mean of the target variable in the training set. The interpretation of $R^2_{Holdout}$—while similar in some respects to the standard in-sample $R^2$—has important differences. A value of 1 still indicates perfect prediction. A value of 0 suggests the model's predictions on the holdout set are no better than simply using the training set mean (or prevalence in the case of binary dependent variables). Negative values are possible in contrast to standard $R^2$, indicating that the model's predictions on new data are worse than using the training set mean. Unlike the standard $R^2$—which is always

between 0 and 1 for models with an intercept—the $R^2_{Holdout}$ can take on any value less than or equal to 1 (i.e., be infinitely negative in the case of infinitely inaccurate predictions). This reflects its nature as a measure of out-of-sample predictive performance, rather than in-sample fit. The $R^2_{Holdout}$ provides a more realistic assessment of a model's predictive capability on new, unseen data, making it particularly useful for evaluating and comparing predictive models. It's worth noting that our implementation—which uses cross-validation—technically results in what[57] refer to as $R^2_{CV}$. While $R^2_{Holdout}$ and $R^2_{CV}$ are conceptually similar in that they both assess out-of-sample performance, they have slightly different estimands. $R^2_{CV}$ provides an estimate of expected performance on a new sample from the same population, while $R^2_{Holdout}$ estimates performance on a specific held-out dataset. While we use the term $R^2_{Holdout}$ throughout this paper for consistency with previous literature in our field, we take this chance to acknowledge the technical distinction between $R^2_{Holdout}$ and $R^2_{CV}$ here.

### Reporting summary

Further information on research design is available in the Nature Portfolio Reporting Summary linked to this article.

## Results

### Essays, teacher assessments and genetic markers predict cognitive and non-cognitive traits

We denote predictive power using $R^2_{Holdout}$ scores, where 0 equals the performance of a prediction based on the training-sample mean, and 1 is a perfect prediction, roughly comparable to the well-known and more traditional in-sample $R^2$; see Section Methods for details regarding our approach and our supplementary material for full details. With this metric, the performance of NLP-based information sets from essays for cognitive abilities and non-cognitive traits approaches or exceeds teacher assessments (TA, see Fig. 2). Predictions are highest for reading (at age 11, NLP = 0.59; TA = 0.57; at age 16 NLP = 0.58; TA = 0.56), verbal (0.55; 0.57), mathematical (at age 11, 0.55; 0.57; at age 16, 0.55; 0.62) abilities and smaller for non-verbal ability (0.37; 0.45). Genetic predictions are smaller, with 0.14 for reading ability at age 11, 0.15 for reading ability at age 16, 0.13 for verbal ability at age 11, 0.16 for mathematical ability at age 11, 0.17 for mathematical ability at age 16, and 0.11 for non-verbal ability at age 11. The prediction for non-cognitive traits is much less precise in general. For occupational aspiration, essays have of $R^2_{Holdout}$ of 0.11, TA 0.11, and PGSs of 0.04. The pattern is similar for scholastic motivation (0.08; 0.09; 0.05), whilst externalizing shows a moderate prediction based on TA (0.08; 0.19; 0.04), and internalizing behavior is least predictable (0.03, 0.08, and 0.01). Additionally, we found that our models could predict 'Big 5' personality traits measured at age 50, with essays predicting over 10% of the variance in agreeableness and openness (see Supplementary Section Predicting Personality at Age 50 and Supplementary Fig. S10). As outlined in Section Methods, we used L1 regularization to pre-screen variables in each fold of the SuperLearner.

### Embeddings drive performance of textual prediction

We aimed to understand the efficacy of our textual prediction model by decomposing our SuperLearner(s) into a series of discrete submodels. These submodels consisted of: a) Traditional readability-metrics, b) Orthographic and grammatical error ratios, c) Advanced computational linguistic measures, including a total of 566 metrics relating to lexical characteristics and sentiment, and d) 1536-dimension, deep-learning-based embeddings. We also further decomposed the individual sets of variables that are extracted from the textual information. The results—as shown in Fig. 3—were illuminating. The comprehensive model outperforms the essay length alone as a predictive benchmark by a factor often exceeding 5–10 times, clearly showing the strength of more intricate feature engineering over simple, text-based proxies. However, the incremental utility when incorporating the full model to one solely employing the embeddings presents only a marginal enhancement, indicating that almost all information presented above is incorporated within them.

### Combining essays, teacher assessments, and PGS maximizes the prediction of educational attainment and cognitive ability

Figure 4 depicts predictions for a general factor of cognitive ability at age 11 as well as educational attainment. For cognitive ability, accuracy obtained by TA is the highest of all three baseline models at 0.62, closely followed by NLP at 0.57, with PGSs at 0.15.

We also calculated $R^2$ metrics for simple models that contain only the first ten genetic Principal Components. See the (in-sample) results for an ordinarily linear regression in S2, and (out-of-sample) results in a Super-Learner framework in S3. The results are indicative of the fact that the Principal Components have no explanatory power, which is not surprising given the homogeneity of the sample. Indeed, adding them to the existing set of scores does not change predictive results beyond decreasing the ratio of signal to noise, as shown in S4.

A further question concerns the added benefit of combining all three types of data studied so far in a single model: Combining TA and NLP improves predictive ability by 0.11. The additional incorporation of PGSs improves the prediction for both TA and DL only minimally. When all three approaches are combined into an ensemble model, the mean accuracy across folds increases to 0.70, demonstrating the benefit of using multiple sources of information. For educational attainment, TA shows the highest mean accuracy (0.29), followed by NLP (0.26) and PGS (0.19). Combining TA and the NLP prediction only marginally improves the TA by 0.07, but additional information from the teacher improves the NLP-based results even further. PGSs prediction adds to TA (0.24) and DL prediction (0.28). Incrementally, in comparison to the two other approaches, each of TA (0.19), NLP (0.06), and PGSs (0.23) adds predictive power through their own unique information content.

### Essays, teacher assessments, and PGSs predict educational attainment comparably to cognitive skills and non-cognitive traits

The joint overall SuperLearner prediction with a model built on each of the three main data sources (TA, NLP, PGS) for educational attainment at age 33 is 0.38, superseding linear models based on cognitive ability, as well as non-cognitive traits (Fig. 5). That analysis also includes birth weight and height as two popular biological predictors of education for comparison. However, those measures only predict attained education with an accuracy of 0.01 and 0.03, respectively. To put the performance of our models into context, one of the most heralded sociological predictors of educational attainment—parental education—is only able to predict with an accuracy of 0.12. Importantly, we rebuild an extended version of the sociological model from the FFC to predict educational attainment—in parallel to their outcome measure of GPA—based on the child's sex at birth, father's education level, mother's education level, social class of the mother's husband, household living density (persons per room), father's socioeconomic classification and social class of the mother's father. These results from our more sociological model act as a vaguely proximal replication of that earlier work, with an $R^2_{Holdout}$ of 0.18 in a generealised linear framework (0.19 with a SuperLearner).

## Discussion

We utilize recent developments in NLP to re-estimate how accurate social and psycho-social prediction can be based on the extent to which small amounts of text can predict the cognitive skills, non-cognitive traits, and educational attainment of individuals. Our study shows—in contrast to previous claims[9,37,58]—that significant predictions of such outcomes are possible. Predictions for educational attainment nearly double the predictive accuracy seen in the more temporally proximal measured Grade Point Average or Material Hardship of the FFC. Our text-based predictions reach an accuracy of 0.56 based on a shorter text (~250 words) and in a smaller, yet more representative sample than the most closely related existing research, which leverages textual data[24]. The prediction of our best model approaches the test-retest reliability of benchmark intelligence tests[59]. While our predictions for non-cognitive traits compare favorably to the benchmark predictions for 'grit' in the FFC, these traits appear to be less easily predicted

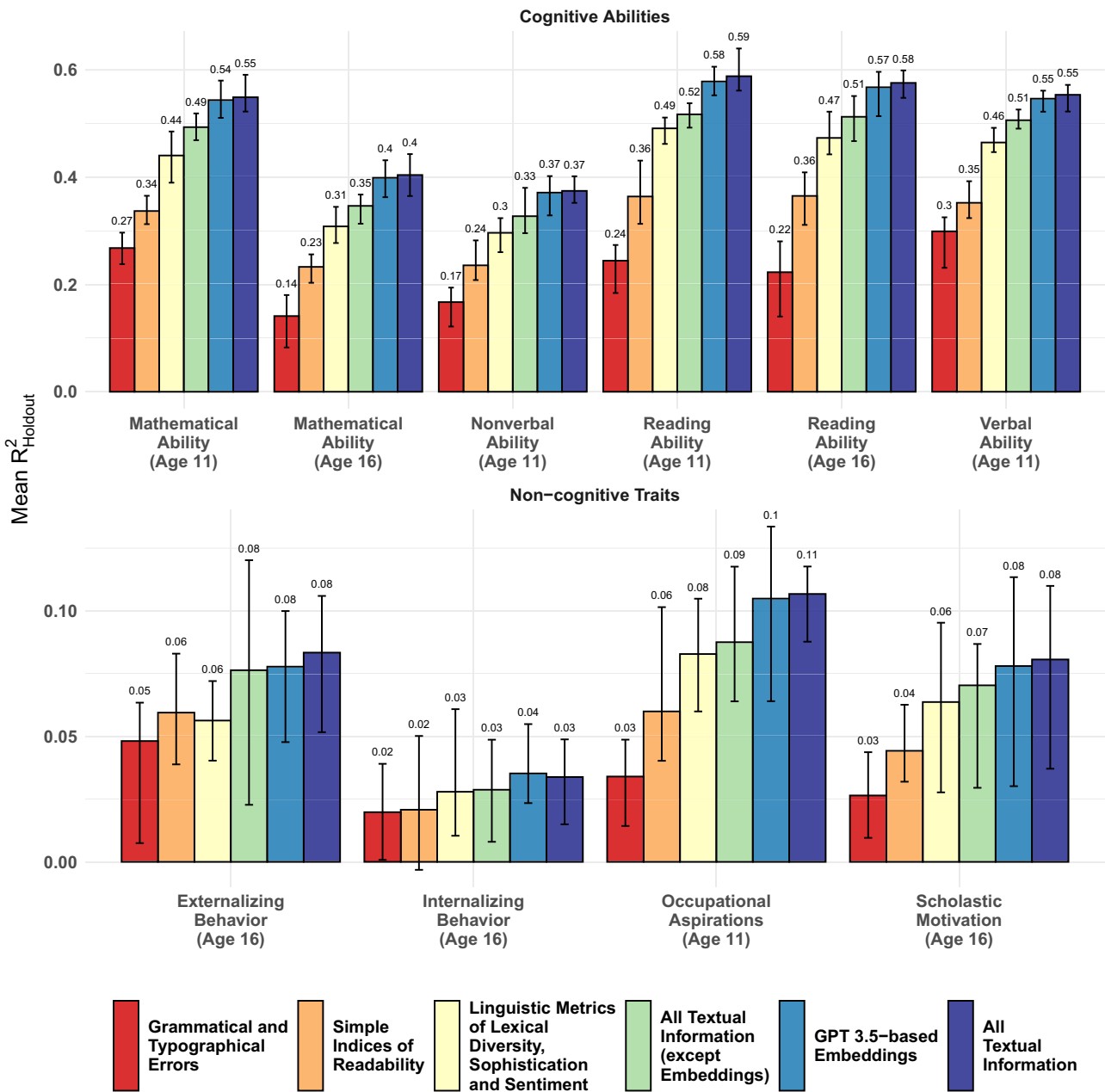

**Fig. 3 | Performance of different components of textual prediction.** Whiskers mark the lowest and highest $R^2_{Holdout}$ over all CV-Folds, with the bars representing the mean $R^2_{Holdout}$. Note: the bars are not cumulative, but independent.

than cognition and education based on our data and methodology. The FFC provided participants with information on both test scores and a teacher survey, one of our main predictors. Heterogeneity in prediction success can therefore have other sources, too, such as sample selection, the exact intricacies of the measurement of outcomes of interest, and the homogeneity of participants, natural to the birth cohort study, which we use in contrast to other data sources.

Predictions of psychological and social outcomes are highly relevant to various current debates. While survey data might be limited in its ability to provide precise predictions, our study shows the opportunities available from alternative data and, in the process, identifies space for positive intervention. Our approach offers another perspective on this 'gloomy prospect'[18,37] of unpredictablity. Additionally, the juxtaposition of "man" against "machine" has a long tradition[60], and we show how progress in the field produces reasonable essay assessments as a function of NLP algorythms[61]. In this sense, AI-based automation of teaching tasks is a potential challenge in current

debates on technology and education[62], with growing interest dedicated to the task of automating essay scoring[63]. Will algorithms such as those developed within this paper one day replace the role of teachers' assessments, or help to re-frame the debate around alternative sources of admissions data? We show that—based on the *current* (and at the time of publication arguably already outdated) technologies—teachers' assessments still provide significant predictive accuracy, and we subsequently stress the importance of expert opinion at a time when some look to automation. However, given that textual samples were able to provide an almost equivalent predictive power to the teacher evaluation-based predictions, there is clear evidence that machines can support teachers' work in schools too, with further methodological development surely forthcoming in this area.

In the past, textual data has been of rather limited utility to quantitative and computational researchers. Although our approach presents an example of how information encoded within complex data resources can be utilized, it also puts into perspective the use of prediction based on genetics

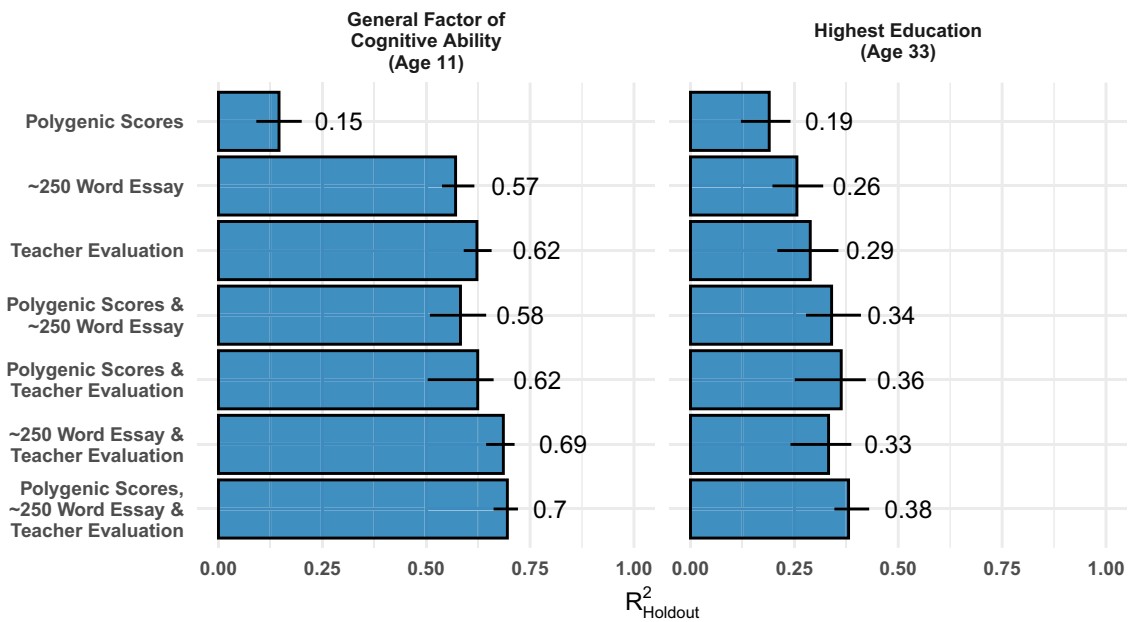

Fig. 4 | **Prediction of highest attained education and general cognitive ability at age 33 combining essays, teacher assessments, and PGS compared to each of the three baseline predictions ($N = 3399$).** Whiskers mark the lowest and highest $R^2_{Holdout}$ over all CV-Folds, with the bars representing the mean $R^2_{Holdout}$.

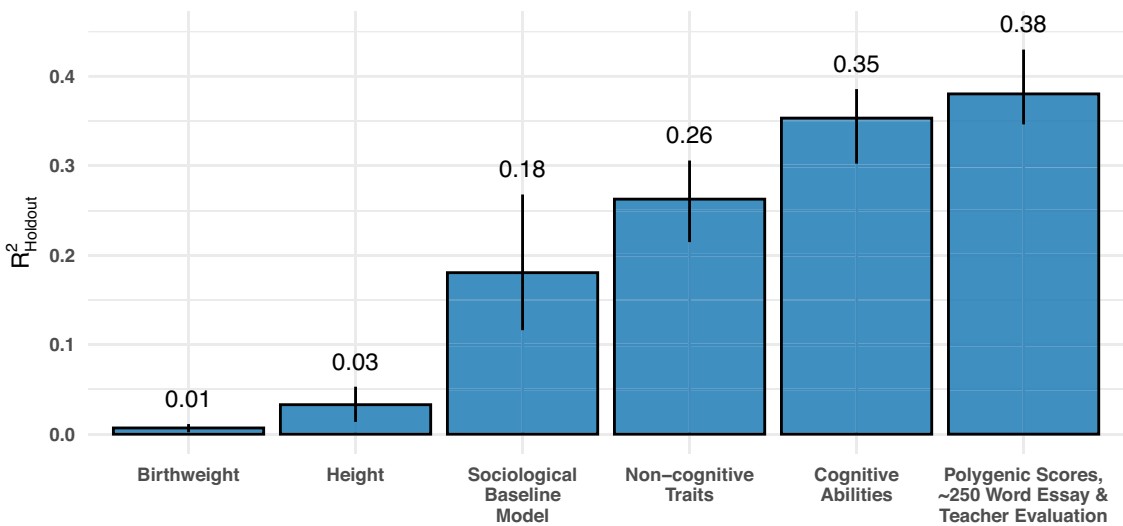

Fig. 5 | **A model containing all three information sets (TA, NLP, PGSs) estimated with a SuperLearner-based approach is compared to generalized linear models based on cognitive ability factor, non-cognitive traits, parental education, birth weight and height.** The outcome of interest is the highest attained education at age 33. For sample sizes, kindly see Table S6. Error bars mark the lowest and highest $R^2_{Holdout}$ over all folds, where the height of the bars represents the mean $R^2_{Holdout}$.

for collegiate admissions[40] in comparison to other sources of complex information, although we acknowledge the potential to further improve in the future in conjunction with ever more powerful genetic discoveries[64]. It must also be emphasized that our results should be contextualized within a specific place, space and time. We also note the cost—in terms of both data collection and computational processing—of working with textual and genomic data. For that reason, in limited resource settings, we show and emphasize here the value that remains in the use of teacher assessments; they perform well, and are otherwise relatively inexpensive to collect and process.

As alluded to in Section Introduction, this work is only possible because of the design of one specific and rather unique dataset, which by definition prohibits exploration of external validity. While showing promise with respect to its predictive performance and a clear need for the increased collection of non-standard data such as this (the expense of doing so not withstanding), our work also raises ethical questions, as it will almost certainly be the case that predictive accuracy continues to increase across and

be integrated into all aspects of society. Confidence in applied predictive systems has led to mistakes in the past, not least in the fields of recidivism[65] and credit default[66]. Bias in educational testing also has a storied history, anticipating many aspects of the modern algorithmic bias and fairness literature[67], and is none more important than within the era of large language models. As digital measures are slowly emerging, identifying the appropriate regulation of increasingly accurate applications of (social and behavioral) prediction remains a consideration for further ethical research.

## Limitations

Although we demonstrate the predictability of psycho-social outcomes, our study is not without limitations. First, it is unclear to what extent our results are generalizable as they are based on a particular sample of individuals born in the year 1958 in Great Britain, and their educational biographies might not be comparable to those of today's contemporary pupils within or outside of the UK. This underscores the need for more longitudinal datasets that

combine different and diverse types of non-standard data with classical social surveys and behavioral assessments. In our supplementary information (Section Additional Analyzes), we address potential concerns related to variations in sample composition due to different non-response patterns across different variables. Specifically, we reran our primary analyses on a subset of 1618 individuals with full information across all key variables. Although predictive performance showed increased variation in this reduced sample, our findings remained robust. Despite these robustness checks, we acknowledge the potential influence of systematic missingness—especially in the genetic subsample—on our results.

Second, prediction alone is not sufficient to understand the factors that drive the association between non-standard data sources and outcomes, and future research should investigate the mechanisms connecting the information in the essays with the successful prediction of the outcomes.

Finally, we have not data mined the three sources of information as thoroughly as we might have done, or utilized more emergent GPT models. An example of this is the fact that we created PGS using Clumping and Thresholding with a prespecified $p$ value threshold of 0.5, and we want to make our applications as standardized as possible; custom thresholds per trait would likely increase the predictive accuracy of our models even further. The same holds for more sophisticated Bayesian methods of PGS construction that leverage functional information about the genome[68], which may further increase predictive performance, though likely not qualitatively change results. Similarly—and while we found no discernible difference between the models which we employed—Large Language Models are currently evolving rapidly, and analysis done on closed models is subject to change at the behest of those who release their trained models. See Section Methods for additional information on the variations we have explored in general.

## Data availability

All non-genetic data associated with the project can be downloaded from the website of the UK Data Service, organized in sets of .dta files, a joint one for Sweeps 0–3 and additional ones for each subsequent wave. Essays are accessible separately. Access to genetic data requires a separate application to the data access committee of the Center for Longitudinal Studies (CLS) and was obtained by the authors under application GDAC-2021-13-TROPF. Our embeddings are available upon request for the purposes of replication to accredited parties who have fulfilled all necessary compliance-related steps.

## Code availability

The code (acknowledging data access restrictions) is available at github.com/tobiaswolfram/llm_paper.

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

## Acknowledgements
The author is deeply indebted to Charles Rahal and Felix C. Tropf. This manuscript has been presented at the Social Science Genetics Network Conference 2022 in Bologna and at the International Society for Intelligence Research Conference 2023 in Berkeley. A preprint is available at https://osf.io/preprints/socarxiv/a8ht9.

## Funding

## Competing interests
The author is an employee of a pre-launch company working in the genomics sector.
