## [Transparent Peer Review file · Communications Psychology]

Large Language Models Predict Cognition and Education Close to or Better than Genomics or Expert Assessment

Corresponding Author: Dr Tobias Wolfram

Version 0:

Decision Letter:

Dear Mr Wolfram,

Thank you for your patience during the peer-review process. We apologize for the delay in reaching a decision. Your manuscript titled "Large Language Models Predict Cognition and Education Close to or Better than Genomics or Expert Assessment" has now been seen by 4 reviewers, and I include their comments at the end of this message. They find your work of interest but raised some important points. We are interested in the possibility of publishing your study in Communications Psychology, but would like to consider your responses to these concerns and assess a revised manuscript before we make a final decision on publication.

We therefore invite you to revise and resubmit your manuscript, along with a point-by-point response to the reviewers. Please highlight all changes in the manuscript text file.

Editorially, we consider it important that you provide greater clarity on your methods and results. The reviewers point to numerous places where the details of the methods, analyses, choice of terminology, and results are imprecise or the rationale for the methodological or analytical decision is absent or unclear (e.g., how embeddings are extracted and configured into the model, hyperparameters of LASSO, explanation on R2, dataset not clearly presented on the figures, LDPred-based PGS, etc.). We consider it important that your revision focus on providing concise, yet precise, descriptions of your methods and results.

Please prepare your code and data so that they can be peer-reviewed when you submit your revision. These should be provided in such a way that reviewers can anonymously access both.

Given the potentially sensitive nature of your study, we asked Reviewer 3 to complete a Risk Assessment. You will find this assessment at the bottom of this letter. It is not necessary for you to respond to this assessment in your point-by-point response to reviewers.

I am attaching an Editorial Requests Table that details critical reporting requirements for the revised manuscript. Please attend to each item and ensure your manuscript is fully compliant. We are requesting that your manuscript aligns with these requirements as this facilitates the evaluation of your manuscript, reducing delays in re-review and potential future acceptance. If your revised manuscript is not aligned with these requests on major issues, such as those concerning statistics, it may be returned to you for further revisions without re-review. Additional information can be found in our style and formatting guide <https://www.nature.com/documents/commspsychol-style-formatting-guide-accept.pdf> Communications Psychology formatting guide.

Please use the following link to submit your

- revised manuscript,
- point-by-point response to the referees' comments,
- cover letter (as a separate document),

- the Editorial Policy Checklist (see below),
- the Reporting Summary (see below), and
- the completed Editorial Request Table (attached):

Link Redacted

Best regards,

Jennifer Bellingtier

Jennifer Bellingtier, PhD
Senior Editor
Communications Psychology

Jixing Li, PhD
Editorial Board Member
Communications Psychology
orcid.org/0000-0002-5210-6224

REVIEWER EXPERTISE:

Reviewer #1 machine learning, LLM
Reviewer #2 cognitive/educational attainment
Reviewer #3 longitudinal studies, computational social science
Reviewer #4 genomics

REVIEWER REPORTS:

Reviewer #1 (Remarks to the Author):

This paper uses a variety of computational tools and techniques to show how large language models can be used to accurately predict cognition and educational outcomes (which the title communicates effectively). The paper is novel and interesting in terms of data, ideas, methods, and findings. I do believe however the paper could be strengthened by adding more clarity and making the arguments stronger and more direct. However, based on the quality of the current version I also believe this authorship team can make these adjustments. Please see the attached document for my specific notes and comments.

Also, I am aware that some of my comments/notes are not addressable with the format of the journal. If that is the case, please indicate so in your response.

Reviewer #2 (Remarks to the Author):

This paper leverages a unique longitudinal dataset to explore the extent to which cognitive abilities, non-cognitive traits and socioeconomic outcomes in adult life can be predicted using a) teacher assessment at age 11, b) genetic variability c) measures derived from an essay produced by children at age 11 (importantly using recent developments in Natural Language Processing).

My own expertise as a reviewer lies in a deep understanding of the dataset being used, a longstanding interest in life course analyses, experience of using the sample of essays for both teaching and research (but not using NLP), and a real interest in the development of NLP in social science research.

The paper clearly makes a major contribution to scholarship and appears to be rigorous both in documenting the variables used and the models and techniques used for the analysis. It is very interesting that the authors situate their research in comparison with the Fragile Families Challenge which was much less successful in predicting outcomes even though a wealth of data was available.

It is also commendable that this jointly authored paper is able to present analyses that leverage sophisticated GWAS alongside cutting-edge use of natural language processing and more conventional survey data.

Section 2.2 and the associated figure is particularly interesting, but I needed to read it several times to fully understand what is being argued here. Further discussion of why the deep-learning based embeddings are so successful as predictors is needed and some type of exemplar to help the non-technical reader to understand the nature of these embeddings would be really helpful.

Where the paper is less successful is in engaging a more general reader, clearly communicating the main results of the analysis, and emphasising what is most novel about the paper. This may be appropriate given the technical and scholarly nature of the journal, but ideally more attention could be paid to the way that results are presented and discussed so that the paper achieves the reach and impact it deserves.

More specific points:

Given that the paper is focussed on the power of these individual essays to predict later life outcomes it is a real shame not to provide the reader with some examples of the essays or even with references to previous work that has analysed the essays using more hermeneutic qualitative approaches. The reader is repeatedly reminded that the essays are approximately 250 words in length, but it is only in Figure D1 that the reader really gets a sense of the heterogeneity in terms of length of the essays. The difficulties of transcribing handwritten essays and the number of spelling mistakes etc is also obscured. More on this could be included in section A1 but also at least some mention in the main text given that the essays are at the heart of this analysis.

Although the supplementary figures provide very useful contextual information on the distribution of key variables (e.g. essay length) they do not include sample sizes which makes it difficult for the reader to be sure whether the distributions presented are for the maximum sample within the dataset (or the restricted sample that had all variables available for analysis and modelling e.g. missing listwise).

The supplementary Tables in section E could also be better annotated to make them more readily understandable.

Reviewer #3 (Remarks to the Author):

In this manuscript the authors measure the predictability of different life outcomes (e.g. reading ability, occupational aspirations, externalizing behavior, etc) using different kinds of data (genetic data, teacher evaluation, 250 word essay). Some of the outcomes occur at age 11 (the same time that the non-genetic data is measured), some outcomes occur at age 16, some occur at age 30, and some occur at age 50 (in the appendix, not the paper). The authors claim to find high levels of predictability for some tasks and low levels of predictability for other tasks (although the abstract seems to focus most on the high levels of predictability).

Overall, I think this paper covers an interesting and important set of issues. I thank the authors for their contribution. There is a lot going on in this paper, and I mean that in both a good and bad way. In a good way, I mean that they combine different data sources and outcomes to explore a range of predictability measures. In a bad way, I mean that it is sometimes hard for me to follow, and I wish the authors had expressed their findings more clearly (at least to me) and synthesized the results more. Overall, I think these changes would both increase the impact of the manuscript. I do not think the paper is publishable in its current form, but I think it could be publishable---and impactful---with further revisions.

I will divide my comments into major (which I think really need to be addressed, but of course I would defer to the editor) and minor (which I think can be left up to the authors and editor). They are presented in a numbered list in the hopes of being easier for the authors.

Major concerns/comments/questions/issues (in no particular order):

1) I think the title is inaccurate and misleading. I don't think it is factually correct to say that the large language models "predict" anything in this paper. The predictions are all being done by SuperLearner. Instead, some of the features come from large language models. I also think the title is misleading in the sense that the SuperLearning predicts *some* cognitive and educational outcomes between that genomics or expert assessment. This does not seem to be true for educational attainment at age 33 for example (Fig 4). Could the authors please explain the choice of title?

2) An important missing piece in the manuscript is a clear consideration of time. Some of the more striking claims in the abstract come from using data measured at birth or age 11 to predict outcomes at age 11. Calling this a prediction is misleading to me, even though I fully acknowledge that the term is used this way by some in the field. At a minimum, it needs to be clear what is a contemporaneous prediction and what is a future prediction. This could be illustrated clearly in Fig 1 and described more clearly throughout the text.

3) The authors use terms like "predictive ceilings" but they have no way of knowing if these are ceilings or not, nor is this term defined clearly. For example, as described below, I don't think they even used all the available data in their dataset.

4) If I understand correctly there is substantially more data in the NCDS (e.g., parent's level of education). This data does not seem to be used. This choice seems surprising since they compare with FFC, where such survey data was used. I think that their choice could be a reasonable, but I'd like to understand why they made it. This seems like an important design decision that should be explained. To be clear, I'm not asking for a change, just an acknowledgement and an explanation. More generally, I would like to better understand why they picked the outcomes and predictors they did, and how their findings relate to all the other prediction tasks possible with this data.

5) Perhaps I missed it, but there are few details about how the analysis that was conducted that I think should be included (probably in the appendix). 1) How was hyper-parameter tuning done, if at all? 2) How many features were kept/dropped in

the LASSO preprocessing and was that preprocessing done once or once per outcome or once per outcome fold or some other way?

6) I did not understand Fig 3. Can these results be presented in terms of R^2 (as with other figures) rather than percent increase? It is hard to tell if these are large relative changes for very small numbers (e.g., an increase from 0.01 to 0.05).

7) I did not understand the multiple mediation model. I did not understand the findings or the assumptions underlying the model. The only paper they site is an unpublished working paper from 2017. What this model is attempting to do seems extremely difficult. Without more context, I'm very skeptical. Can the authors explain the method more completely and/or justify their confidence in these conclusions? To be clear, I'm not saying the model is wrong or used incorrectly. I'm saying that I don't understand it and I'm skeptical of the results.

8) I liked the analysis in SI Section C.4. It seems like the results from other parts of SI Sec C are not mentioned in the paper, even though I think they are important. For example, why was predicting personality at age 50 in the appendix instead of the regular paper? These results appear to be qualitatively different, with much lower levels of predictability. More generally, could the authors explain why some of the results in SI Section C were not mentioned in the main paper (or sorry if I missed them).

Minor concerns/comments/questions/issues (in no particular order):

1) In the results section, I find it confusion to compare "deep learning" and "teacher assessment". Deep learning is a class of models like say random forest, but teacher assessment is a kind of data. To me the more natural comparison is between data sources. Also, it is not even clear to me if this is really deep learning on the essay, since it seems to be to use a pre-trained model to get features and then use SuperLearner. I would like to see more clarity in language.

2) The second paragraph of this paper has a lot going on. Have the authors considered adding a new paragraph break between the results of the FFC, which did not use text, and the results from Alvero et al., which does use text. If one of the main innovations of this paper is the use of the essay (which seems to be claimed in the introduction), then making this distinction clearer might help. Also, the discussion of genetic data feels just stuck on the end and not really developed. Again, maybe another paragraph might help. More generally, it seems like the authors are setting up a "horserace" between essay, teacher evaluation, and genetics. Perhaps one paragraph about each? Obviously, this choice should be left to the authors.

3) I didn't understand the paragraph on line 88 – 99. I don't see how this is related to this paper.

4) On line 109, I don't see how the results of the FFC suggests that only genes are useful for prediction nor that genes should be used for college admissions. The FFC does not even have genetic data. This claim should be clarified or removed.

5) On line 116, I was not clear what is meant by "Heterogeneity in predictions"

6) Around page 3 and 4 I think it would be very helpful to have a schematic (like Fig1) but describing the NCDS data.

7) I really liked Fig 1. I think it would be more helpful if it included information about the time of the outcomes (e.g., age 11, age 16, etc).

8) I found it hard to distinguish between the colors in Fig 2. This makes it hard to interpret one of the main results of the paper.

9) The authors say they used R^2_{holdout} but I think they are really using R^2_{CV} . I think the difference in meaningful because the estimand for R^2_{CV} is different (see Bates et al. 2023 <https://www.tandfonline.com/doi/full/10.1080/01621459.2023.2197686>). To be clear, I'm not asking for a change in metric, just a clarification (or perhaps I'm misunderstanding the analysis).

10) I find it confusing that the R^2 values are presented as percentages, which does not match the equation in Section 4.2.

11) I don't understand the use of color shading in Fig 4.

12) In Section 2.3, I don't understand what is meant by an ensemble model. Did you include all the predictors at once or did you do some kind of averaging of the predictors from each model?

13) In Fig 5, I didn't understand what was changing and what as staying the same. Is there a change in the model class (linear models vs superlearner), predictors, or both. As a reader I was expecting only one thing to change at a time, but the authors might have a different goal.

14) I don't understand the part of the conclusion related to automation anxiety. Are they claiming that feature representations from essays might replace teacher assessment? If so, they should make this claim explicitly and justify it. If not, they should rework this section to avoid confusion.

15) I'm curious how the authors decided to order and group the results in Section 2.

16) The authors claim that they used a state-of-the-art pretrained model, but this model was released in December 2022 (<https://openai.com/blog/new-and-improved-embedding-model>), and there have been big improvements since then. I could see a few options for how to deal with this. The authors could remove the claim about "state-of-art" or time restrict it. It might also be interested to explore how the results change with newer models. My guess is that they would not change by much, but if they did it might be evidence that our ability to predict with text will continue to improve as models get bigger/better. Since prediction with text is so key to the framing of the paper, this should be addressed in some way, I think.

17) The authors claim on page 12 that SuperLearner "guarantees" that they can extract the "maximum predictive validity from the three data sources". I don't think SuperLearning makes any such guarantees in finite sample size (what they have here), but I do think it is a reasonable choice. I would suggest that the authors clarify this claim or remove it.

18) Does the SuperLearner provide the weights for the weighted average that it uses? Could the authors report in the appendix, which models get most heavily weighted for which outcomes? These results seems like something that they already have and it might be useful in future research.

I would like to include a note to the editor and authors:

1) I am not an expert in genetic data, and I am not able to review that part of the analysis.

I would like to conclude by thanking the authors again for their interesting manuscript.

Reviewer #4 (Remarks to the Author):

The manuscript provides a valuable addition to the literature describing the advantages of including polygenic scores in multi-modal datasets for prediction and the extent of their capabilities compared to other data resources. Moreover, the authors include related social commentary, albeit only briefly, by questioning the usefulness of genomics in future policy making. A subject not without controversy that needs the academic community to be more prominently featured in the debate in any context. Its inclusion is therefore appreciated.

Throughout the manuscript the authors use the word 'genes'. When referring to genotyped data, this term is incorrect since the information contained in the data goes down to single nucleotide polymorphisms (SNPs). Hence, it is necessary to refer to the genetic data of the NCDS sample as e.g., 'genetic markers' or 'SNPs'. When referring to polygenic scores, 'genetic predisposition' or 'genetic indices' for a certain phenotype (e.g., educational attainment, is more appropriate whenever neither their abbreviation ('PGS') nor their full name ('polygenic scores') are used. In titles, tables, and figures 'PGS' is the most appropriate and convenient term to use.

The paper by Becker et al. (2021) in the footnote on page 12 needs to be properly cited and added to the 'References' section. The authors mention in the same footnote having obtained similar results when using LDpred-based PGS on preliminary analysis. Replication of results based on different methods strengthens the credibility of the paper's analyses and main conclusion regarding PGS. The manuscript would thus benefit from describing this in a bit more detail. The authors should consider expanding briefly on these additional analyses in supplementary sections.

The authors describe in section A.3 of the supplement the construction strategy of the PGS. While the fact that this data has been quality controlled and imputed is acknowledged, further details are not provided. Providing this additional information is advised since it would increase the work's methodological transparency and would aid future replication attempts. The list of useful details entails but is not limited to the filters used to exclude low quality SNPs/samples and the imputation pipeline (if applicable, imputation servers used, well-established imputation workflows such as RICOPII and settings therein such as filters according to imputation quality scores and MAF, version of the reference genome, etc). In the case that the above steps were not (entirely) carried out by the authors themselves since this is a dataset used in past research by many different groups, a reference that documented all details should be provided.

The p-value threshold for SNP-inclusion in the PRSice algorithm was set to $p=0.5$. To set a single threshold might not be ideal given that the authors use summary statistics from a wide range of phenotypes due to the varying genetic architecture across traits. Consequently, the predictive power of these genetic measures is not well exploited. That is, the largest possible amount of variance of the outcome variables explained by the PGS is most likely not reached with the current settings. The authors could achieve more predictive genetic features by performing 'best-fit PGS' scoring already implemented in PRSice on each and every single summary statistics trait. The idea is to produce PGS at multiple thresholds, run linear regression with each PGS on the outcome variables and pick the p-value threshold at which the corresponding PGS explain the most variance. Finally, use the best-fit PGS to rerun the main analyses described in the main sections. Statistical overfitting represents no issue here since the authors apply nested cross-validation to control for it. The construction of the PGS used in the main analyses needs further clarification. The authors mention 33 different traits, yielding 33 different PGS but for the main analyses the reader is led to interpret that just one PGS feature was included. Was a composite measure utilized for the main analyses (e.g., a mean/sum score across all 33 PGS) or were all PGS entered independently and yet simultaneously to the SuperLearner analyses? If the latter is true, as the multiple-PGS approach cited in lines 139 to 140 would suggest, then why is just a single value reported on all main plots (and the range of variation indicated by the whiskers) and described in the results section? A short summary of the multi-PGS approach is needed around those same lines and/or in later sections.

The issue of (genetic) population stratification needs to be addressed since no mention of it is apparent in the manuscript. Associations between PGS and phenotypes are bound to be confounded by genetic similarity among individuals. The use of similarity measures can be included as covariates in the main analyses. Alternatively, before running main analyses, genetic similarity may be regressed out of the PGS via multiple regression. Genetic similarity can be operationalized as principal components extracted from the genotyped data using PLINK's approach to principal component analysis or multidimensional scaling. The number of appropriate components to control for can be determined e.g., by including only components that explain a vast proportion of genetic variance (as determined with a scree-test or with the Kaiser-Gutmann criterion). In the case that population stratification has been dealt with already, its procedure should be stated in the main text.

On table E.1. depression is categorized as a personality trait. Depression is a mental disorder and should be classified alongside ADHD, ASD, BP, and SCZ as a 'Disease'.

Risk/benefit analysis provided by Ref #3

Are there potential risks/dangers with publishing this work?

I don't see any.

Could the methods outlined in this work be misused to pose a threat (for example, to data privacy)?

The methods used in this paper create no “marginal risk” (risk above and beyond the risk that already exists).

The methods used in this paper are not that different from those widely available in machine learning textbooks.

What steps could be taken to mitigate the risks? Specifically, we’d appreciate your input on whether data, code, detailed methodology should be presented so as to minimize risks and adverse impact from reuse.

None.

The data themselves should not be free to download, but the data is already protected by UK Data Service (see also footnote on page 1).

If the code and/or data was not shared publicly (beyond limited academic use), could a reader develop something similar based on the methods in the paper?

Yes.

If the information were to be broadly communicated “as is,” what is the potential for:

1. Public misunderstanding and harm

- What might be the implications of such misunderstandings, e.g., psychological, social, health decisions, economic, commercial etc.?

There might be a potential for misunderstanding as some people may have fears related to using genetic data for prediction. I personally think this work is within the norms of other work using genetic data such as work generating polygenic scores. But, I am not an expert in genetics. You are welcome to ask one of them.

2. Sensationalism

- In what way might it result in widespread concern or even panic about data privacy, contact tracing apps, or other safety/security issues?

I don’t think this will lead to widespread concern or panic about data privacy, contract training apps or safety/security.

There is a small chance that the work might be sensationalized by journalists. I would encourage you to encourage the authors to review the manuscript (and more importantly any press releases) to ensure accuracy and reduce sensationalist claims.

Benefit analysis

Are there potential benefits (for example, to data privacy) from application or utilization of this information? If so, please describe.

Yes, there is benefit from understanding predictability of life outcomes.

Will this information be useful to the scientific community? If so, how?

We do not know what outcome are predicable from what data using what methods. This is an important part of understanding the life course.

Risk vs. Benefit Assessment

Based on the risks and benefits identified, and considering the

time frame in which these might be realized:

- Do the benefits of communicating the information outweigh the risks?

- Do the risks outweigh the benefits?

In my opinion the benefits outweigh the risks.

That said, I would encourage you to encourage the authors to review their manuscript with these questions in mind, and then keep them in mind if they do outreach to the press.

EDITORIAL POLICIES

We ask that you ensure your manuscript complies with our editorial policies and reporting requirements.

To that end, we require revised manuscripts to be accompanied by two completed items: a reporting summary that collects information on study design and procedure, and an editorial policy checklist that verifies compliance with all required editorial policies.

- <https://www.nature.com/documents/nr-reporting-summary.zip>>Nature Research Reporting Summary
- <https://www.nature.com/documents/nr-editorial-policy-checklist.pdf>>Editorial Policy Checklist

All points on the policy checklist must be addressed. Your revised manuscript can only be sent back to the referees if these checklists are completed and uploaded with the revision.

Notes: If you have submitted a Stage 1 Registered Report, Review, Primer, Comment, or Perspective you do not need to submit these forms. If you have already submitted these forms, you may disregard this request.

If you experience problems in linking your ORCID, please contact the <http://platformsupport.nature.com/>>Platform Support Helpdesk.

Version 1:

Decision Letter:

Dear Dr Wolfram,

Thank you for your patience during the peer-review process. Your manuscript titled "Large Language Models Predict Cognition and Education Close to or Better than Genomics or Expert Assessment" has now been seen by 4 reviewers, and I include their comments at the end of this message. They find your work of interest but raised some important points. We are

interested in the possibility of publishing your study in Communications Psychology, but would like to consider your responses to these concerns and assess a revised manuscript before we make a final decision on publication.

We therefore invite you to revise and resubmit your manuscript, along with a point-by-point response to the reviewers. Please highlight all changes in the manuscript text file.

Editorially, we consider it important that the revised manuscript addresses Reviewer 4's request for a sensitivity analyses that compare the results in the presence and absence of statistical control of genetic ancestry.

I am attaching an Editorial Requests Table that details critical reporting requirements for the revised manuscript. Please attend to each item and ensure your manuscript is fully compliant. We are requesting that your manuscript aligns with these requirements as this facilitates the evaluation of your manuscript, reducing delays in re-review and potential future acceptance. If your revised manuscript is not aligned with these requests on major issues, such as those concerning statistics, it may be returned to you for further revisions without re-review. Additional information can be found in our style and formatting guide <https://www.nature.com/documents/commspsychol-style-formatting-guide-accept.pdf> Communications Psychology formatting guide.

Please use the following link to submit your

- revised manuscript,
- point-by-point response to the referees' comments,
- cover letter (as a separate document),
- the Editorial Policy Checklist (see below),
- the Reporting Summary (see below), and
- the completed Editorial Request Table (attached):

Link Redacted

Best regards,

Jennifer Bellingtier

Jennifer Bellingtier, PhD
Senior Editor
Communications Psychology

on behalf of
Jixing Li, PhD
Editorial Board Member
Communications Psychology
orcid.org/0000-0002-5210-6224

REVIEWER EXPERTISE:

Reviewer #1 machine learning, LLM
Reviewer #2 cognitive/educational attainment
Reviewer #3 longitudinal studies, computational social science
Reviewer #4 genomics

REVIEWER REPORTS:

Reviewer #1 (Remarks to the Author):

The authors have gone far above and beyond in addressing my original comments along with those from the other reviewers.

At this point in the process, I only have one additional comment regarding the methodology of this paper. Closed LLMs like the GPT series can be altered behind the scenes without our knowledge at any time, so I do worry about replicability. The authors partially address this by using multiple models, but in the final version I would implore the authors consider potential ramifications of their findings if, for any reason, the LLMs were to change in such a way that their results would not be replicable. At this stage I would not recommend any additional analyses, but the authors have a unique opportunity to sketch out these implications, and any insights would be valuable.

Beyond this I recommend this paper for publication.

Reviewer #2 (Remarks to the Author):

The paper has clearly been extensively and carefully revised in response to reviewers' detailed comments on the manuscript. The argument in the paper and the results are now much easier for a non specialist to follow and I believe that this will enhance the significance of the paper.

I am not a specialist on NLP or genetic testing so I am not able to comment on these aspects of the paper but my in-depth knowledge of the 1958 cohort study gives me confidence that the authors have done an excellent job in leveraging the detailed longitudinal and multi-disciplinary data within the study. The detail in the paper is also likely to inspire further useful rigorous work that extends and expands the current analysis using different sets of variables, different outcome variables and increasingly sophisticated NLP.

Reviewer #3 (Remarks to the Author):

I would like to thank the authors for their improved manuscript. I believe that they sufficiently address the concerns of the reviewers, and I believe the paper should be published.

I appreciated several of the improvements made by the authors such as adding more information about time (e.g., Fig 1), clarifying some analysis (e.g., Fig 3), and adding some new analysis (e.g., Fig D.3). I also liked the way they dealt with the fact that the GPT has and will continue to change over time; I found it interesting that there was little gain from GPT 3.5 to GPT 4.

At this point I offer only suggestions/comments that I would fully leave to the authors. It is not my role as a reviewer to write the paper, and it will have the authors names on it (not my own).

- I continue to think the title is misleading for the reasons described in my first review. I didn't find the authors' comments convincing, but this is their paper with their names on it.
- I thank the authors for their improved attention to time in the text and figures, but I personally would still do more for the reasons described in my initial review.
- I really like Fig 3, but the first time I read it I misunderstood it. The first time I read it I thought that the bars were cumulative in the sense that the second bar (simple indices of readability) includes that variable and previous variables (grammatical and typographical errors). I'm not sure why I had this misunderstanding or if it is worth trying to fix.
- My favorite part of the paper was section 2 (results). I felt like the section 1 (introduction) and section 3 (discussion) sometimes ranged far beyond what was in the paper and in a way that was not needed.
- The limitations section is important but a bit hard to parse now. There are many different things all packed into one paragraph. I wonder if they could group these into say a few categories and then signpost them more clearly (e.g., First, . . . Second, . . . Finally . . .) I know the authors have thought a lot about the limitations, which I appreciate, and I wish their deep insights could be communicated more easily to the readers.
- I think it is interesting that Fig 3 seems to show that 1) we get more predictive accuracy from GPT 3.5 embeddings than any human extract features (e.g., index of readability) and 2) adding all human extracted features to the embeddings seems to add no predictability. I wonder if the authors think this is a general pattern or whether it might be specific to this dataset.

Again, I offer these pieces of feedback with the hope that they help the authors. Ultimately, I would leave it to them to decide.

Overall, I think this is an interesting paper, and that it was improved by this round of revisions.

Reviewer #4 (Remarks to the Author):

I thank the authors for addressing the raised concerns pertaining the previous manuscript submission with such care. For the most part, adequate changes were introduced to the revised version, either by supplying additional information or by providing reasonable arguments to support analytical choices. However, there is one major point in need of more thorough examination, alongside a few very minor ones, the latter being concerned merely with formalities in the way information is

reported/displayed. As such, the following is divided into “major” und “minor” comments. Kindly also consider the references added as a third segment of my response that correspond to the indices placed throughout the comments.

Major comment:

1) In the revised manuscript, as well as in the response to the previous comments, the authors take the informed decision to abstain from assessing and mitigating the effect of population stratification in analyses involving PGSs. The standpoint to support this decision proposes that genetic variation stemming from genetic ancestry is a legitimate source of PGS prediction. While this sounds reasonable since PGS are constructed based on the very fact that people vary in their allele frequencies in all polymorphic loci (which is determined to a substantial degree by ancestry), information contained in genetic load on traits are dissociable from that contained in genetic ancestry. Between closely related individuals (e.g., siblings) inherited genotypes will not be identical since they arise due to random independent meiotic events, which are not influenced by population stratification¹. It is this portion of signal, as opposed to that shaped by genetic ancestry, that is the subject of academic research that tries to optimize genetic scores for prediction and possibly for future disease risk assessment in health care settings. Leaving population stratification uncontrolled for exacerbates the already existing problem of interpreting what PGS truly measure. Work by David Curtis² in 2018 serves as an example. The study concluded that both magnitude and distribution of Schizophrenia PGS were highly dependent on ancestry. Individuals of African descent had a 10-fold higher genetic risk for schizophrenia (as measured with PGSs) than European individuals, when the PGS were constructed using GWAS summary statistics derived from European samples. In all likelihood, this does not mean that people of African genetic background are at higher risk, since the difference in PGSs between individuals with- and without a schizophrenia diagnosis was still significantly different from 0 after controlling for population stratification and since the disease prevalence is roughly the same across populations worldwide. Furthermore, the PGS relationship with ancestry has been repeatedly found to reduce greatly the already weak predictive performance of PRS, when GWAS summary statistics are applied for PGS construction to genotyped data from individuals that do not match the genetic ancestry of the discovery GWAS³. The PGSs' performance varies even between individuals of the same ancestry⁴, though to a smaller degree. Thus, biased performance estimates due to this kind of confounding is very likely without stratification correction/control in the present manuscript since the GWAS summary statistics used were obtained from studies with populations of varying ancestry. It should be noted, however, that since the population under investigation in the present manuscript is exclusively British, the impact of confounding due to ancestry is likely to be less than in multi-ancestry or non-European studies. Nevertheless, not negligible.

As the authors point out in their response to my comments, confounding is of interest in studies aiming at causal inference, but as laid out above, this applies for studies focusing on prediction as well. I therefore request the report of sensitivity analyses that compare the results in the presence and absence of statistical control of genetic ancestry. As mentioned in the last revision round, principal components derived from PLINK's “—pca” command offer a good solution, but the choice of technique to operationalize this construct is of course left to the authors. I recommend including the first few principal components (e.g., four, or an empirically supported number. For the latter, see recommendations on the last revision round) as covariates in PGS models, but reporting the fraction of total explained variance that is solely attributable to the PGS (if programmatically feasible, otherwise, kindly refer to the last round of revisions for other alternatives).

Finally, the authors express their concern for performance drops after ancestry control. Regardless of potential performance drops, the manuscript's scientific value could benefit from this procedure because the interpretability of the PGSs and, by extension, of the results, would become easier and alternative explanations questioning the results' validity would be ruled out.

Minor comments:

1) In the footnote on page 15, while the typesetting error was removed, the paper itself is still not cited correctly. It should read “[...] the extensive set of polygenic scores provided by the Polygenic Index Repository (Becker et al., 2021), constructed using LDpred2 (Privé et al., 2020), was also tested in preliminary analyses [...]”

2) According to the footnote on page 15, the method is ‘LDpred2’ but on line 915, ‘LDpred’ is mentioned. Please check which of both methods was used.

3) Figure D.1 of the supplements: This addition is much appreciated. Figure readability could be improved by making sure that the text of the holdout mean R² annotations don't overlap.

4) Table E.1 of the supplements: As per my comment during the first round of revisions, depression should fall under the category “disease” rather than “personality trait”.

References

- 1 Brumpton, B., Sanderson, E., Heilbron, K., Hartwig, F. P., Harrison, S., Vie, G. Å., ... & Davies, N. M. (2020). Avoiding dynastic, assortative mating, and population stratification biases in Mendelian randomization through within-family analyses. *Nature communications*, 11(1), 1-13.
- 2 Curtis, D. (2018). Polygenic risk score for schizophrenia is more strongly associated with ancestry than with schizophrenia. *Psychiatric genetics*, 28(5), 85-89.
- 3 Duncan, L., Shen, H., Gelaye, B., Meijssen, J., Ressler, K., Feldman, M., ... & Domingue, B. (2019). Analysis of polygenic risk score usage and performance in diverse human populations. *Nature communications*, 10(1), 3328.
- 4 Ding, Y., Hou, K., Xu, Z., Pimplaskar, A., Petter, E., Boulier, K., ... & Pasaniuc, B. (2023). Polygenic scoring accuracy varies across the genetic ancestry continuum. *Nature*, 618(7966), 774-781.

EDITORIAL POLICIES

We ask that you ensure your manuscript complies with our editorial policies and reporting requirements.

To that end, we require revised manuscripts to be accompanied by two completed items: a reporting summary that collects information on study design and procedure, and an editorial policy checklist that verifies compliance with all required editorial policies.

- <https://www.nature.com/documents/nr-reporting-summary.zip>>Nature Research Reporting Summary
- <https://www.nature.com/documents/nr-editorial-policy-checklist.pdf>>Editorial Policy Checklist

All points on the policy checklist must be addressed. Your revised manuscript can only be sent back to the referees if these checklists are completed and uploaded with the revision.

Notes: If you have submitted a Stage 1 Registered Report, Review, Primer, Comment, or Perspective you do not need to submit these forms. If you have already submitted these forms, you may disregard this request.

Communications Psychology is committed to improving transparency in authorship. As part of our efforts in this direction, we are now requesting that all authors identified as 'corresponding author' create and link their Open Researcher and Contributor Identifier (ORCID) with their account on the Manuscript Tracking System prior to acceptance. ORCID helps the scientific community achieve unambiguous attribution of all scholarly contributions. You can create and link your ORCID from the home page of the Manuscript Tracking System by clicking on 'Modify my Springer Nature account' and following the instructions in the link below. Please also inform all co-authors that they can add their ORCID to their accounts and that they must do so prior to acceptance.

If you experience problems in linking your ORCID, please contact the <http://platformsupport.nature.com/> Platform Support Helpdesk.

Version 2:

Decision Letter:

Dear Dr Wolfram,

Your manuscript titled "Large Language Models Predict Cognition and Education Close to or Better than Genomics or Expert Assessment" has now been seen by our reviewer, whose comments appear below. In light of their advice I am delighted to say that we are happy, in principle, to publish a suitably revised version in Communications Psychology.

We therefore invite you to revise your paper one last time to address the remaining concerns of our reviewers and a list of editorial requests. At the same time we ask that you edit your manuscript to comply with our format requirements and to maximise the accessibility and therefore the impact of your work.

EDITORIAL REQUESTS:

Please review our specific editorial comments and requests regarding your manuscript in the attached "Editorial Requests Table". Please outline your response to each request in the right hand column. Please upload the completed table with your

manuscript files as a Related Manuscript file.

SUBMISSION INFORMATION:

OPEN ACCESS:

* DATA AVAILABILITY:

Link Redacted

Best regards,

Jennifer Bellingtier

Jennifer Bellingtier, PhD
Senior Editor
Communications Psychology

Jixing Li, PhD
Editorial Board Member
Communications Psychology
orcid.org/0000-0002-5210-6224

REVIEWER EXPERTISE:

Reviewer #4 genomics

REVIEWERS' COMMENTS:

Reviewer #4 (Remarks to the Author):

I want to highlight the author's attention to the smallest of details and issues, a testament of their professionalism. Their dedication devoted to my past comments is no exception.

The effort put by the authors to address the effect of population stratification in the sensitivity analyses is much appreciated. I think conclusions drawn from them provide valuable additional information that increases the validity of the results reported in the main text. As the authors point out in their answers to my past comments, these analyses show indeed a lack of predictive power of the genetic principal components alone and imply an inconsequential impact of population stratification on the PGSs' predictive capabilities in the sample at hand. However, the principal components are meant to be handled as covariates to the rest of predictors of interest (i.e., the PGS), and not as the only regressors present in the predictive models since their unique contribution to prediction performance is of less interest. According to footnote 2 of section 2.3, the sensitivity analyses were carried out doing the latter.

These covariates are to be included alongside the PGS in the statistical models of the sensitivity analyses. The performance reported should be that of the PGS within those models or, if not programmatically possible, of the whole model. The supplementary analyses reported in Tables E2 and E3 should be updated to reflect these changes. Additionally, for the sake of clarity, the title of supplementary table E3 should explicitly refer to the usage of PGSs and ten principal components in the SuperLearner models.

This way, the predictive performance of the PGSs can be clearly contrasted between models with- (in a revised version the supplementary tables) and without (as already reported in the main text without any need of further changes) population stratification control.

After this update, I firmly believe that the manuscript will be suitable for publication.

We further delineate hereafter the changes which we have made to the appropriate parts of the manuscript in response to the all of the insightful comments of each reviewer in turn.

Comments From Reviewer One

Kindly note: the first part of this response relates to the text returned to us outside of the annotated .pdf file.

Reviewer One Comment 1 (R1.1): *This paper uses a variety of computational tools and techniques to show how large language models can be used to accurately predict cognition and educational outcomes (which the title communicates effectively). The paper is novel and interesting in terms of data, ideas, methods, and findings. I do believe however the paper could be strengthened by adding more clarity and making the arguments stronger and more direct. However, based on the quality of the current version I also believe this authorship team can make these adjustments. Please see the attached document for my specific notes and comments.*

Author's Response: We sincerely thank the reviewer for both their enthusiasm and their comments, and are delighted that they found utility within our work. The main suggestion provided as text (as opposed to annotation on the attached pdf) is a relatively recurring one: We have edited the manuscript throughout to add clarity and directness. We have, for example, also considered Reviewer Three's comments as appropriate so as to not make our revisions overly strong, and ensure in the process that we are not making overly hyperbolic statements. Please also see here our reply to comment E1.

Kindly note: this next part of our response relates to the annotations in the pdf.

Reviewer One Comment 2 (R1.2): *Machines can generate human like (written) language, but "understanding" is overselling it, especially when framing this particular paper.*

Author's Response: Yes, we entirely agree, and apologise for using overly strong language in a previous version of the manuscript. We have corrected the wording from *understanding* to *interacting* with humans in accordance to the issue raised by the Reviewer.

Reviewer One Comment 3 (R1.3): *Why is this important though? Isn't there some kind of trade-off between interpretability and accuracy/power here that makes the advantages not so obvious?*

Author's Response: We thank the reviewer for this important point. While we acknowledge the potential trade-off between interpretability and accuracy of prediction models, our approach serves the key purpose of advancing predictive capabilities by identifying advances in computationally advanced text-based prediction. In the process, we provide a novel comparison between three distinct predictor classes: NLP-based predictions, teacher assessments, and genomics. However, we also discuss the capability of

machines to mimic human behavior and therefore we believe that our contribution remains distinct from this issue; the design of all of our Figures in the main body of the text – for example (i.e. Figures 2-5) are designed to show the marginal increments of each specific information set. This serves, we feel, in a way, to give substantive interpretability with regards to exactly what is powering our results. See revised Figure 3, for example: indicates that the majority of the performance in the 'All Textual Information' category (i.e., all information extracted from essays) is being powered by the GPT-3.5 based embeddings. It would have – in addition – been possible to investigate this matter further, but we feel that the focus is on precision within our study, as opposed to interpretability, and do not want to complicate our delivery by diverging too far into this area.

Reviewer One Comment 4 (R1.4): *This whole section could be greatly strengthened with some more synthesis. The cites/references are all relevant, but it's not clear how they intersect and inform this specific study. Addressing this point and my previous point in the intro could make the rationale and motivation more obvious. **This** book chapter might have some helpful language/framing to consider:*

Author's Response: We here again appreciate the Reviewer's comment, and we have rearranged our introduction entirely accordingly. We appreciate but were not hitherto aware that our study represents an example for the meta-theoretical perspective highlighted by the reviewer, which indeed glues our empirical investigation to social science theory and common intuitions about statistical testing. We therefore conclude our theoretical remarks based on the reference provided and another such example from the literature. It is – in this context – important to highlight that the 'gloomy prospective' of unpredictability and how it represents another example of the meta-theoretical realm which works on deriving 'predictability hypotheses', not least regarding historic events (Risi, 2019), which effectively are testable based on predictive algorithms as opposed to inferential testing (van Loon, 2023).

Reviewer One Comment 5 (R1.5): *I think some simple, formal notation here would be really helpful, like $Y = Xb + e$, etc. etc. For example, how exactly were the word embeddings incorporated into the model? The norm? Some kind of average? I looked in the supplementary materials and couldn't find anything. Definitely not encouraging the authors to include each and every step, but adding some formal scaffolding would go a long way towards helping me/readers make sense of this diagram.*

and

Reviewer One Comment 6 (R1.6): *I was having a hard time identifying the specific 'metrics' used in the text and supplementary material. I saw the TAALED and others, were they all used? How big are these predictive models? And was the text the only data that used the 10-fold CV?*

and

Reviewer One Comment 7 (R1.7): *Same question as above: how were these configured into the models? I'm assuming that the other methods are basically document level measurements for each document, but what was your approach for the embeddings? Further, how exactly did you extract the embeddings? For example, did you identify individual words in the corpus (ie. the vocabulary) and then pull the respective embeddings from this larger model? Or did you fine-tune the model with the essays and then extract those?*

Author's Response: We thank the reviewer for articulating these questions so eloquently regarding our metrics and methodology. We also appreciate the opportunity to clarify these points:

Information Extraction: As detailed in our supplementary material, we employed several approaches to extract information from the essays:

- GPT-based embeddings using the text-embedding-ada-002 model (see also our next reply), resulting in a 1,536-dimensional vector for each essay (i.e., despite the fact that that predictive performance was largely invariant to the GPT 3.5 or 4.0 model used).
- SALAT-metrics, providing 534 measures related to various linguistic aspects.
- Readability-metrics, consisting of 31 different measures from the koRpus-package.
- Grammatical and typographical error ratios, derived using the LanguageTool CLI. With specific regard to the embeddings please see response R1.7.

Size of predictive models: Our main predictive model – as shown in Figure 2 – incorporates different types of text-based features. The model is large and comprehensive, and as detailed above, uses four key types of extracted information. As per Figure 3, we further decompose and now make clearer (see R3.6 and R3.18) the results from the individual components of the model, and in the text demonstrate the incremental utility of each category compared to a simple benchmark using only the length of essays.

Cross-validation: We used 10-fold cross-validation for all three of our models (textual data, genomic data, and teacher assessments). However – as we document in the supplementary material – the performance of a linear model was comparable for genomic data and teacher assessments. The textual data, on the other hand, benefited significantly from the SuperLearner based approach.

In response to these points, we further complement our Material and Methods section. Note that in line with our response to E1, we now make our introduction more accessible, with technical details appearing in our Material and Methods section. We refer early in our Results section to the Material and Methods section and in the latter to the supplementary material for additional, complete details which would permit replication of our study.

Reviewer One Comment 8 (R1.8): *A few questions here:*

1. *What are the baselines being compared?*
2. *Are these R-squareds or some kind of classification (eg. nominal DVs) accuracy?*
3. *If they are classification models, what are the modes/SDs/etc?*

Author's Response: We appreciate the opportunity to clarify these points, and sincerely apologize for any lack of clarity in our original submission. To address your specific questions:

1. **Baselines and metrics:** We use the R^2_{Holdout} metric for all our predictions. This metric compares model performance to a baseline prediction using the mean of the training set. As indicated elsewhere in this report, we have emphasized this for example as our first sentence in our results section: We denote predictive power using R^2_{Holdout} scores (where 0 equals the performance of a prediction based on the training-sample mean, and 1 is a perfect prediction: kindly see Material and Methods for details regarding our approach and our Supplement for a complete set of details).
2. **Nature of the models:** These models predict continuous outcomes, not classifications. In the case of educational attainment, these are utilised as an ordinal variable (see Section B.3).
3. **Variability of results:** As these are regression models, modes and standard deviations aren't directly applicable.

To improve clarity, we have substantially revised Section 4.2 as appropriate.

Reviewer One Comment 9 (R1.9): *What were these again (text-based proxies)?*

Author's Response: Kindly see our response to the reviewers clarification request R1.5 above for all details.

Reviewer One Comment 10 (R1.10): *Aren't there other comparisons I should be making here? Based on the intro and abstract, it seems like the thrust of this paper is about how computational tools can outperform traditional approaches; are those approaches represented here?*

Author’s Response: We agree with the reviewer’s comment and consider it an excellent idea to engage in a direct comparison with the sociological baseline model for prediction as articulated in the Fragile Family Challenge (FFC). We have done so with a generalised linear model which includes child’s sex at birth, father’s education level, mother’s education level, social class of the mother’s husband, household living density (persons per room), father’s socioeconomic classification and social class of the mother’s father, and have added the results of this more sociological model to Figure 5 in the main text next to other popular predictors such as height and birth weight. Very comforting, our results (0.18 in a generalised linear model, 0.19 in a SuperLearner) are almost identical with the prediction of GPA in the FFC, and we now highlight this in the abstract.

Our model therefore outperforms the standard sociological model in reference to the literature, drastically and dramatically further strengthening our results. We updated the results and methods section accordingly. We thank the reviewer for this valuable input.

Reviewer One Comment 11 (R1.11): *Adding some context with the actual text would go a really long way, like showing what exactly is being picked up by the GPT 3.5 embeddings. Adding text snippets/excerpts could be one way to do that.*

Author’s Response: We entirely agree with the Reviewer and our revised analysis of how various textual features are predicting cognitive and non-cognitive abilities presented in Figure 3 hopefully gives better insights into the information which is meaningful for the essay-based prediction. We are looking forward to more in depth work being done in this area in the future. We do most sincerely agree that it would furthermore be insightful to provide more details about the essays as well as, for example, text snippets. However, due to various information compliance and governance restrictions, we are prohibited from sharing the raw the content of the essays. We are also unsure as to whether we would be able to share extractions or embeddings from our models, either, and note that that information alone in a high-dimensional space might not necessarily aid with regards to other comments received regarding the accessibility and ‘directness’ of the paper.

Reviewer One Comment 12 (R1.12): *Same question as before: what are the dimensions/inputs/IVs/etc. for these models? Since R-squared increases with more variables, adding this context could be helpful in interpreting these results*

Author’s Response: We hope that we have been able to answer the questions pertaining to the dimensionality and inputs in earlier parts of this response. We would quickly note that our pseudo-R² metric is an *out-of-sample* metric, independent of *in-sample* fit; when building statistical or machine learning models for prediction, the inclusion of more variables does not necessarily lead to better out-of-sample fit. We have clarified details of the metric used in Section 4.2 as described in our response to comment R1.8.

Reviewer One Comment 13 (R1.13): *How exactly though? I could buy this claim but I'm not 100% sure how to make sense of the connection between this and the primary findings of the study*

Author's Response: We appreciate that the link between our work and the discussion here was perhaps previously slightly too tangential. We now insert a hypothetical question into this question to aid the linkage, and explicated and emphasize our point in the process (kindly see Section 3). Additionally, the juxtaposition of “man” against “machine” has a long tradition, and we show how progress in the fields produce reasonable assessments of the ability of NLP algorithms. In this sense, AI-based automation of teaching is a potential challenge in current debates on technology and education, with growing interest dedicated to the task of automating essay scoring. Will algorithms such as those developed within this paper one day replace the role of teachers' assessments, or help to re-frame the debate around alternative sources of admissions data?

Reviewer One Comment 14 (R1.14): *In my opinion, I feel like these results surprised me the most. They were relatively simple yet seemed to be the pound-for-pound strongest predictors. Although these might not be the main point of the paper I still think there's something to say about simpler vs. more complex approaches, relative advantages, etc. Like, where in the paper is there a sentence along the lines of "although the teacher evals worked well, these computational approaches are still preferable for reasons A, B, C with respect to education, X, Y, Z, etc.". I don't know what those would be but I think adding in that kind of language would really enhance the paper.*

Author's Response: We find this point extremely insightful and valuable. It speaks especially to the fact that both textual and genomic data are expensive to curate and process. In comparison, teacher assessments represent comparatively good ‘pound-for-pound’ value, as the reviewer so expertly puts it. To this extent, we have now incorporated a mention of this important fact into the penultimate and final sections of the discussion section:

“We also note the cost – in terms of both data collection and computational processing – of working with textual and genomic data. For that reason, in limited resource settings, we show and emphasize here the value that remains in the use of teacher assessments; they perform well, and are otherwise relatively inexpensive to collect and process.”

Reviewer One Comment 15 (R1.15): *The Alvero et al. paper cited in the intro also used readability scoring and noted that they performed poorly. Were any of these metrics strongly correlated with any of the outcomes?*

Author's Response: In response to Reviewer R3.6, we have substantially improved the ease of interpretation for Figure 3, which now includes better annotations as to exactly which of the individual metrics (in terms of readability scoring) are generating which

degrees of predictive accuracy. This compares directly to the readability of the GPT-3.5 model, and a joint model of all readability metrics we combined (kindly also see Material and Methods, Section 4, for further detail). While it would be possible to incorporate this information (i.e. the mean, max, and minimum held-out R2) into a table, we would strongly prefer to stick to this Figure as a presentation of our findings in the text – as our message is directed towards a comparison with other text-based predictions – and this is well facilitated by the gradual increment of the bars.

Reviewer One Comment 16 (R1.16): *This approach was cool, reminded me of **this** paper.*

Author’s Response: We had not previously been aware of this insightful paper, and now cite it alongside Alvero et al. (2020). We do already have a citation of Steven Jones (2013), but – despite the similarity in names – they coincidentally appear to not be the same author!

Comments From Reviewer Two

Reviewer Two Comment 1 (R2.1): *The paper clearly makes a major contribution to scholarship and appears to be rigorous both in documenting the variables used and the models and techniques used for the analysis. It is very interesting that the authors situate their research in comparison with the Fragile Families Challenge which was much less successful in predicting outcomes even though a wealth of data was available. It is also commendable that this jointly authored paper is able to present analyses that leverage sophisticated GWAS alongside cutting-edge use of natural language processing and more conventional survey data. Section 2.2 and the associated figure is particularly interesting, but I needed to read it several times to fully understand what is being argued here.*

Author's Response: We are delighted with this comment from Reviewer Two, especially with regards to our careful positioning of our work with the Fragile Families Challenge (FFC), which was very much a deliberate objective of ours. We rewrote the introduction and methods section and made amendments to the entire manuscript hopefully resulting in a clearer build-up to and narrative within Section 2.2. We now also include – as alluded to in our response to R1 – a more directly comparable sociological model which has almost equivalent performance as to the baseline GPA predictions in the FFC.

Reviewer Two Comment 2 (R2.2): *Further discussion of why the deep-learning based embeddings are so successful as predictors is needed and some type of exemplar to help the non-technical reader to understand the nature of these embeddings would be really helpful.*

and

Reviewer Two Comment 3 (R2.3): *Where the paper is less successful is in engaging a more general reader, clearly communicating the main results of the analysis, and emphasising what is most novel about the paper. This may be appropriate given the technical and scholarly nature of the journal, but ideally more attention could be paid to the way that results are presented and discussed so that the paper achieves the reach and impact it deserves.*

Author's Response: These are recurring remarks made also from the Editor (see comment E1) and Reviewer One (see comment R1.6-8). We have therefore entirely revised our Introduction and Methods sections, and worked through the presentation of our findings and discussion a little more clearly. For some details, kindly also see our explicit responses to E1 and R1.6-8 with regards to those suggestion.

Reviewer Two Comment 4 (R2.4): *Given that the paper is focussed on the power of these individual essays to predict later life outcomes it is a real shame not to provide the reader with some examples of the essays or even with references to previous work that has analysed the essays using more hermeneutic qualitative approaches. The reader is repeatedly reminded that the essays are approximately 250 words in length, but it is only in Figure D1 that the reader really gets a sense of the heterogeneity in terms of length of the essays. The difficulties of transcribing handwritten essays and the number of spelling mistakes etc is also obscured. More on this could be included in section A1 but also at least some mention in the main text given that the essays are at the heart of this analysis. Although the supplementary figures provide very useful contextual information on the distribution of key variables (e.g. essay length) they do not include sample sizes which makes it difficult for the reader to be sure whether the distributions presented are for the maximum sample within the dataset (or the restricted sample that had all variables available for analysis and modelling e.g. missing listwise).*

Author's Response: We entirely agree with the Reviewer and R1.11 which are similar points. Our analysis of various textual features predicting abilities presented in Figure 3 hopefully gives some insights into the information which is meaningful and complementary to constructs such as the word embeddings. We are looking forward to more in depth work here in the future. As above, we agree that it would be informative to provide more details about the essays in the form of, for example, text snippets. However, due to various information compliance and governance considerations, we are prohibited from sharing the content of the essays. Finally, we have added sample size information to all descriptions in the Supplement.

Reviewer Two Comment 5 (R2.5): *The supplementary Tables in section E could also be better annotated to make them more readily understandable.*

Author's Response: We have added more detailed annotations in all Tables and Figures presented in the Supplement, and in the main body of the text we now annotate all figures directly with the metric's mean value.

Comments From Reviewer Three

Reviewer Three Major Comment 1 (R3.1): *I think the title is inaccurate and misleading. I don't think it is factually correct to say that the large language models "predict" anything in this paper. The predictions are all being done by SuperLearner. Instead, some of the features come from large language models. I also think the title is misleading in the sense that the SuperLearning predicts *some* cognitive and educational outcomes between that geonomics or expert assessment. This does not seem to be true for educational attainment at age 33 for example (Fig 4). Could the authors please explain the choice of title?*

Author's Response: We appreciate the Reviewer's concern and would like to clarify our rationale for the title. While the SuperLearner algorithm performs the final predictions, the Large Language Model (LLM) plays a crucial role in our approach. The LLM generates text embeddings that serve as key input features for the SuperLearner, capturing rich semantic information from the essays that enables effective prediction. To address this point, we have restructured Figure 3 to clearly demonstrate that the additional gain from all textual features beyond the LLM-based embeddings is limited. In fact, for almost all outcomes, the SuperLearner using only the embeddings as input performs nearly on par with the full model that includes all textual information. In the context of our approach, the SuperLearner primarily serves as an ensembling tool. While it's beyond the scope of our current study, we believe that an LLM fine-tuned to essay-outcome pairs could potentially achieve similar or even better performance. Regarding predictive performance, both Figure 2 and Figure 4 show that the essay-based SuperLearner's performance is comparable to teacher assessments for virtually all traits, with the exception of externalizing behavior. The overlap in fold results across most cases further supports this observation. Given these considerations, we believe the current title accurately reflects the central role of LLMs in our predictive approach while acknowledging the ensemble nature of our final model.

Reviewer Three Major Comment 2 (R3.2): *An important missing piece in the manuscript is a clear consideration of time. Some of the more striking claims in the abstract come from using data measured at birth or age 11 to predict outcomes at age 11. Calling this a prediction is misleading to me, even though I fully acknowledge that the term is used this way by some in the field. At a minimum, it needs to be clear what is a contemporaneous prediction and what is a future prediction. This could be illustrated clearly in Fig 1 and described more clearly throughout the text.*

Author's Response: We appreciate the attention to the temporal aspects of our predictions and agree that clarity in terminology is crucial. While the term "prediction" is commonly used for both contemporaneous and future outcomes, as noted by the reviewer, we understand how it could potentially be unclear to readers. However, it is important to highlight that our study includes a range of predictions across different time spans:

- Our cognitive abilities predictions at age 11 show strong results, but we also predict cognitive abilities at age 16, demonstrating the model’s ability to predict outcomes 5 years into the future.
- Educational attainment predictions are made for outcomes at age 33, which is 22 years after the age 11 assessments.
- In the Supplement, we include Big 5 personality predictions for age 50, which is 39 years after the initial assessments at age 11. These align with our other non-cognitive trait predictions.

Nevertheless, we added the following sentence to the introduction:

“We examine predictions across various outcomes and time spans, from contemporaneous outcomes such as cognitive abilities at age 11 – when the text samples are collected – to those decades later such as educational attainment of the children at age 33.”

We have also modified Figure 1 accordingly as suggested.

Reviewer Three Major Comment 3 (R3.3): *The authors use terms like “predictive ceilings” but they have no way of knowing if these are ceilings or not, nor is this term defined clearly. For example, as described below, I don’t think they even used all the available data in their dataset.*

Author’s Response: We thank the reviewer for their comment and have revised the manuscript accordingly. Conceptually, we can never in theory know if we are at the ceiling or not, unless we are predicting something perfectly. We have refined our language to be more consistent with what we mean, i.e. the current limits of predictive accuracy as found in the literature.

Reviewer Three Major Comment 4 (R3.4): *If I understand correctly there is substantially more data in the NCDS (e.g., parent’s level of education). This data does not seem to be used. This choice seems surprising since they compare with FFC, where such survey data was used. I think that their choice could be a reasonable, but I’d like to understand why they made it. This seems like an important design decision that should be explained. To be clear, I’m not asking for a change, just an acknowledgement and an explanation. More generally, I would like to better understand why they picked the outcomes and predictors they did, and how their findings relate to all the other prediction tasks possible with this data.*

Author’s Response: We found this recurring comment across reviewers extraordinarily valuable, and in accordance with the this comment and R1.10, we have added an additional model which makes a cleaner comparison within Figure 5, namely that it now includes an equivalent version of the FFC’s sociological baseline model model. This model – which includes child’s sex at birth, father’s education level, mother’s education level, social class of the mother’s husband, household living density (persons per room), father’s socioeconomic classification and social class of the mother’s father – which has been estimated in a linear framework, has been shown to be as predictive as the majority of the ‘common task’ of the 160 teams in the Fragile Families Challenge. Our results nearly perfectly mirror the results therein. Note that the goal of our study is not to maximize the prediction from survey data, but to analyse the comparative performance of text-based predictions.

We further highlight this in the abstract now:

“Combining text, genetic markers, and teacher assessments into an ensemble model, we can nearly perfectly predict cognitive ability (70%, close to test-retest reliability) and explain 38% of individual differences in attainment, whilst a rebuilt of the sociological model which is equivalent to the baseline of the Fragile Family Challenge is able to replicate their relatively low accuracy.”

We again thank the reviewer for their insights which we believe improve our work.

Reviewer Three Major Comment 5 (R3.5): *Perhaps I missed it, but there are few details about how the analysis that was conducted that I think should be included (probably in the appendix). 1) How was hyper-parameter tuning done, if at all? 2) How many features were kept/dropped in the LASSO pre-processing and was that pre-processing done once or once per outcome or once per outcome fold or some other way?*

Author’s Response: We appreciate the opportunity to clarify these important points.

We deliberately chose not to perform algorithm-specific hyper-parameter tuning. This decision was based on several considerations:

- Computational efficiency: Given the large scale of our dataset and the number of outcomes we were predicting, extensive hyper-parameter tuning for each algorithm would have been computationally intensive.
- Avoiding overfitting: By using default settings, we reduced the risk of overfitting that can occur with extensive hyper-parameter tuning, especially when working with a large number of predictors and outcomes.
- Generalizability: Default settings often provide robust performance across a wide range of datasets. Our approach thus tests the generalizability of these algorithms without tailoring them to our specific data.
- Focus on ensemble performance: Our primary interest was in the overall performance of the SuperLearner ensemble, rather than optimizing individual algorithms.

Concerning the LASSO pre-processing: The LASSO pre-processing was indeed performed, and we apologize for not providing more details about this step. To address your specific questions:

- The number of features kept/dropped varied depending on the outcome variable and the specific fold in the cross-validation process.
- The LASSO pre-processing was done once per outcome per fold. This means that for each outcome variable, in each fold of the cross-validation, a separate LASSO model was fit to select relevant features.

We accordingly modified the materials and methods section: In the subsection describing the SuperLearner algorithm, we added:

“We used the default settings provided by the SuperLearner package for each algorithm, without performing additional algorithm-specific hyper-parameter tuning. This approach focused on the overall ensemble performance while maintaining computational efficiency and reducing overfitting risks.”

And briefly afterwards:

“LASSO pre-processing was performed once per outcome per fold in the cross-validation process. For each outcome variable, in each fold, a separate LASSO model was fit to select relevant features. The number of features retained varied depending on the outcome variable and specific fold.”

Furthermore, at the end of section 2.1, we added a footnote on the high-level patterns of feature selection:

“We examined the consistency of feature selection across all folds and outcome variables for each of the three basic SuperLearner models. For the genetic model, all polygenic scores were selected at least once, with household income, educational attainment, and age at first birth being the most consistently selected. In the teacher evaluation model, all covariates were also selected at least once, with general knowledge, number work, and oral ability assessments being selected in every instance. For the essay-based model, while 149 features were never selected, the top 100 most consistently selected features were dominated by embedding dimensions (87) derived from the large language model. Only two non-embedding features from the essays are found among the top 10 most consistently selected: The root type-token ratio and the measure of textual lexical diversity (MTLD), with the former being selected in 96.6% of all models.”

For more details regarding the editing of our Methods (Section 4), please also see our response to R1.7.

Reviewer Three Major Comment 6 (R3.6): *I did not understand Fig 3. Can these results be presented in terms of R^2 (as with other figures) rather than percent increase? It is hard to tell if these are large relative changes for very small numbers (e.g., an increase from 0.01 to 0.05).*

Author's Response: We unreservedly agree that this may have been discontinuous with other parts of our work, and we have modified Figure 3 accordingly to no longer be marginal improvements above the baseline (although we do retain information on these marginal improvements in the main body of the text). Also in response to a later comment from this same reviewer's point (R3.18), we have also generally removed all allusions to percent (of variance explained) to fractal variants of our evaluation metric.

Reviewer Three Major Comment 7 (R3.7): *I did not understand the multiple mediation model. I did not understand the findings or the assumptions underlying the model. The only paper they cite is an unpublished working paper from 2017. What this model is attempting to do seems extremely difficult. Without more context, I'm very skeptical. Can the authors explain the method more completely and/or justify their confidence in these conclusions? To be clear, I'm not saying the model is wrong or used incorrectly. I'm saying that I don't understand it and I'm skeptical of the results.*

Author's Response: We understand the reviewers concern and eventually decided to remove our mediation analysis as the narrative focus of our paper over time has become more about exploring predictive accuracy in a comparative fashion. However, we highlight in the limitations towards the end of the article the need to further explore the mechanistic link between text and the outcomes predicted:

“Prediction alone is not sufficient for understanding the factors that drive the association between non-standard data sources and outcomes. Future research should investigate the mechanisms connecting the information in the essays with the successful prediction of the outcomes.”

Reviewer Three Major Comment 8 (R3.20): *I liked the analysis in SI Section C.4. It seems like the results from other parts of SI Sec C are not mentioned in the paper, even though I think they are important. For example, why was predicting personality at age 50 in the appendix instead of the regular paper? These results appear to be qualitatively different, with much lower levels of predictability. More generally, could the authors explain why some of the results in SI Section C were not mentioned in the main paper (or sorry if I missed them).*

Author's Response:

We thank the reviewer for their interest in our complementary analyses. In accordance with other Reviewer comments, we have decided to drop the mediation analysis and therefore also C.4 from the paper. Future research should investigate in all detail what predictive

information the essays contain, which is not available in survey data. Overall, we utilize all material in the Supplement for our argumentation, largely also within the Supplement itself with cross references. The case of personality is particularly interesting, however, we don't mention it in the main text as this would be yet another angle distracting from our focus on maximizing text prediction for the traits under study and comparing results to those from other predictive sources. Personality is an interesting case for our supplementary analyses as it covers a more distal phenotype to the predictors which is still predictable in general supporting the robustness of our approach.

Reviewer Three Minor Comment 1 (R3.9): *In the results section, I find it confusion to compare “deep learning” and “teacher assesement”. Deep learning is a class of models like say random forest, but teacher assessment is a kind of data. To me the more natural comparison is between data sources. Also, it is not even clear to me if this is really deep learning on the essay, since it seems to be to use a pre-trained model to get features and then use SuperLearner. I would like to see more clarity in language.*

Author's Response: We thank the reviewer for their comment. We have adjusted our language accordingly throughout to make it clear that we are more specifically comparing information sets, and this includes the adjustment of a keyword on the title-page. We have adjusted the acronym used in the results section from 'DL' to 'NLP', which we think better conveys the fact that we are using natural language processing on the information sets as described by the reviewer, and before immediately defining this acronym, we explicate the fact that the NLP tools are used on extracting information from the essays.

Reviewer Three Minor Comment 2 (R3.10): *The second paragraph of this paper has a lot going on. Have the authors considered adding a new paragraph break between the results of the FFC, which did not use text, and the results from Alvero et al., which does use text. If one of the main innovations of this paper is the use of the essay (which seems to be claimed in the introduction), then making this distinction clearer might help. Also, the discussion of genetic data feels just stuck on the end and not really developed. Again, maybe another paragraph might help. More generally, it seems like the authors are setting up a “horserace” between essay, teacher evaluation, and genetics. Perhaps one paragraph about each? Obviously, this choice should be left to the authors.*

Author's Response: Again, many thanks for these insightful comments. As the Reviewer will hopefully appreciate, we have entirely rearranged the introduction to now also include their advice. We hope to have streamlined our narrative, having analytically disentangled paragraphs, and we also clarify the comparisons we are making.

Reviewer Three Minor Comment 3 (R3.11): *I didn't understand the paragraph on line 88 – 99. I don't see how this is related to this paper.*

Author's Response: We have dramatically curtailed this part of our introduction as part of the restructuring of the manuscript as described in other parts of this paper. We do retain a small amount of this content, but instead use it as a motivating factor for our work. This can be found in the fifth paragraph of our revised introduction (Section 1).

Reviewer Three Minor Comment 4 (R3.12): *On line 109, I don't see how the results of the FFC suggests that only genes are useful for prediction nor that genes should be used for college admissions. The FFC does not even have genetic data. This claim should be clarified or removed.*

Author's Response: We have clarified this appropriately by removing the unhelpful and potentially confusing reference to the FFC here. It now reads as follows:

“There are also longstanding questions regarding the role of genetics and the (social) environment, which can be answered by considering the interplay of information generated at differential parts of an individual's life (Dick, 2011). Our work speaks to the ‘gloomy prospective’ of Plomin and others (Plomin, 1987) where it is claimed that only genetic variation (and its phenotypical proxies) are useful for prediction (Plomin, 2011) and – controversially – should be used as information for college admission (Plomin, 2018).”

And we thank the reviewer once more.

Reviewer Three Minor Comment 5 (R3.13): *On line 116, I was not clear what is meant by “Heterogeneity in predictions”*

Author's Response: We have similarly removed this confusing terminology at the appropriate place in the paper.

Reviewer Three Minor Comment 6 (R3.14): *Around page 3 and 4 I think it would be very helpful to have a schematic (like Fig1) but describing the NCDS data.*

Author's Response: We appreciate this suggestion by the reviewer. Instead of providing full details on the birth cohort data, which we consider beyond the scope of this study, we provide references to the full data description of the (NCDS, Power 2006) such as by Power and Elliot which is dedicated to the cohorts description. To make our case of the use of the NCDS, we implemented data links into Figure 1 and the surrounding text which accompanies it (as well as in the supplementary information). We hope this is sufficient to give the reader an impression of our use of the data.

Reviewer Three Minor Comment 7 (R3.15): *I really liked Fig 1. I think it would be more helpful if it included information about the time of the outcomes (e.g., age 11, age 16, etc).*

Author’s Response: Thanks for this comment, we have added the appropriate information as suggested to the revised Figure 1 (and also fixed a typographical error which was apparent within it during the process).

Reviewer Three Minor Comment 8 (R3.16): *I found it hard to distinguish between the colors in Fig 2. This makes it hard to interpret one of the main results of the paper.*

Author’s Response: We increased the contrast in the colours, while retaining our blue colour scheme for consistency throughout the paper. If the Reviewer would prefer, we can adjust this even further and deviate from our colour scheme.

Reviewer Three Minor Comment 9 (R3.17): *The authors say they used R^2_{holdout} but I think they are really using R^2_{CV} . I think the difference is meaningful because the estimand for R^2_{CV} is different (see Bates et al. 2023). To be clear, I’m not asking for a change in metric, just a clarification (or perhaps I’m misunderstanding the analysis).*

Author’s Response: This is a fantastic observation and we have substantially expanded our Analytical Methods section to clarify our terminology. It’s worth noting that our implementation, which uses cross-validation, technically results in what Bates et al. (2023) refer to as R^2_{CV} . While R^2_{Holdout} and R^2_{CV} are conceptually similar in that they both assess out-of-sample performance, they have slightly different estimands. R^2_{CV} instead provides an estimate of expected performance on a new sample from the same population, while R^2_{Holdout} estimates performance on a specific held-out dataset. While we use the term R^2_{Holdout} throughout this paper for consistency with previous literature in our field, we acknowledge the technical distinction between R^2_{Holdout} and R^2_{CV} as described by Bates et al. (2023), and have clarified accordingly (see footnote 2 in Section 4.2.5 for further information).

Reviewer Three Minor Comment 10 (R3.18): *I find it confusing that the R^2 values are presented as percentages, which does not match the equation in Section 4.2.*

Author’s Response: We have corrected this entirely, and now only refer to absolute fractional values of our key metric; please also see our response to point R3.6 (in regards to making Figure 3 absolute values, as opposed to proportional percentage increases in accuracy).

Reviewer Three Minor Comment 11 (R3.19): *I don’t understand the use of color shading in Fig 4.*

Author’s Response: We thank the reviewer for this helpful comment, and appreciate that it was unclear what the shading was in relation to (in the absence of a colourmap); it was – previously – a function of the metric’s value. We have simplified this figure by removing any variation in colour.

Reviewer Three Minor Comment 12 (R3.20): *In Section 2.3, I don't understand what is meant by an ensemble model. Did you include all the predictors at once or did you do some kind of averaging of the predictors from each model?*

Author's Response: Thank you for your question regarding the ensemble model used in our study. We apologize if the explanation in Section 2.3 (and later in Section 4) was unclear. In our study, we employed a machine learning model called a 'SuperLearner' in conjunction with nested cross-validation (CV) to create this ensemble model (as also indicated in Figure 1). The SuperLearner algorithm combines predictions from multiple individual machine learning models, such as Extreme Gradient Boosting, Random Forest, and Support Vector Machines, to generate a final prediction. This is done through a process called 'stacking' or 'model averaging'. The nested CV consists of two loops: an outer loop (10 folds) for model evaluation and an inner loop (10 folds) for model selection. In each iteration of the outer loop, the dataset is divided into training and test sets. The training set is further divided into inner training and validation sets in the inner loop. Within each inner loop iteration, the base learners are trained on the inner training set and make predictions on the inner validation set. The SuperLearner algorithm learns the optimal combination of the base learners' predictions using the inner validation set. This process is repeated for each fold in the inner loop. In the outer loop, the base learners are trained on the entire training set using the settings determined in the inner loop. The SuperLearner algorithm combines the predictions of the base learners using the weights or meta-models learned in the inner loop. The resulting ensemble model is then evaluated on the test set from the outer loop. This process is repeated for each fold in the outer loop, yielding an unbiased estimate of the ensemble's performance. By using nested CV with the SuperLearner, we ensure that the model's performance is evaluated on data that was not used during the model selection process. This helps to prevent overfitting and provides a more reliable estimate of the model's generalization performance.

For general clarification we entirely revised the methods section of our manuscript using some of the content from above (Section 4, and Section 4.2.3. specifically with regards to the SuperLearner), and we now refer to it in the first sentence of the Results section, too. We have also – in response to R3.25 and R3.26 clarified our language on this and included a new figure – Figure D.3 – which visualises the weights of each of the six models which come out of the SuperLearner framework.

Reviewer Three Minor Comment 13 (R3.21): *In Fig 5, I didn't understand what was changing and what was staying the same. Is there a change in the model class (linear models vs superlearner), predictors, or both. As a reader I was expecting only one thing to change at a time, but the authors might have a different goal.*

Author's Response: In Figure 5 specifically, we for illustrative purposes aimed to compare the predictive performance of our SuperLearner ensemble model, which combines

polygenic scores, essays, and teacher evaluations as predictors, to that of linear models using various predictor sets, such as cognitive abilities, non-cognitive traits, birthweight, height, and parental education. This is to better make a comparison of our more sophisticated modeling approach and the non-standard predictors in comparison to something that can be considered to be a more commonly used baseline within the social and behavioral sciences, namely well-known sociological or psychological predictors within a generalised linear framework. We have clarified this – especially in the caption of Figure 5 – and go so far as to note the predictive accuracy of the ‘sociological model’ in the accompanying text (i.e., in the final section of 2.4 which shows that for that model, the SuperLearner increases predictive accuracy by one basis point only).

Reviewer Three Minor Comment 14 (R3.22): *I don't understand the part of the conclusion related to automation anxiety. Are they claiming that feature representations from essays might replace teacher assessment? If so, they should make this claim explicitly and justify it. If not, they should rework this section to avoid confusion.*

Author's Response: We thank the reviewer for pointing out that this was not clear. We are discussing the possibility that ML algorithms replace teaching tasks such as essay correction as our model predicts children's performance. In order to clarify this point, we added the following:

”Additionally, the juxtaposition of “man” against “machine” has a long tradition (Jones, 2013), and we show how progress in the field produces reasonable essay assessments as a function of NLP algorithms (Feigenbaum, 2020). In this sense, AI-based automation of teaching tasks is a potential challenge in current debates on technology and education (Selwyn, 2021), with growing interest dedicated to the task of automating essay scoring (Foltz, 2020)”.

We have furthermore details this more explicitly in accordance with another referee's comment (please see also R1.13 above).

Reviewer Three Minor Comment 15 (R3.23): *I'm curious how the authors decided to order and group the results in Section 2.*

Author's Response: We thank the reviewer for this question about the structure of our results section. We organized Section 2 to present our findings in a logical progression that builds on each previous subsection:

2.1 We begin by establishing the three main predictors (essays, teacher assessments, and genetic markers) and their individual predictive power. This provides a foundation for understanding each predictor's strengths and limitations.

2.2 We then focus on a detailed analysis of the essay-based prediction. We dedicate a separate subsection to this because our essay-based approach, leveraging large language models, shows particularly strong performance and represents a novel contribution to the field.

2.3 Next, we explore how combining these predictors can maximize prediction accuracy. This allows us to demonstrate the synergistic effects of integrating multiple data sources and methods.

2.4 Finally, we contextualize our findings by comparing them with other established predictors in the field. This helps readers understand the relative strength and significance of our approach within the broader landscape of cognitive and educational prediction.

This structure allows us to gradually build a comprehensive picture of our findings, starting from individual predictors and moving towards more complex, integrated models. It also enables readers to clearly see the progression of our analysis and the incremental improvements achieved at each stage.

Reviewer Three Minor Comment 16 (R3.24): *The authors claim that they used a state-of-the-art pre-trained model, but this model was released in December 2022, and there have been big improvements since then. I could see a few options for how to deal with this. The authors could remove the claim about “state-of-art” or time restrict it. It might also be interested to explore how the results change with newer models. My guess is that they would not change by much, but if they did it might be evidence that our ability to predict with text will continue to improve as models get bigger/better. Since prediction with text is so key to the framing of the paper, this should be addressed in some way, I think.*

Author’s Response: We sincerely appreciate this author’s comment. To this extent, we have ran and now compare our results with RoBERTa and GPT 4.0 based embeddings. This is described at the end of Section 4.2.1, and also in Supplementary Figure D.2. The results generally show an invariant performance between GPT 3.5 and 4.0, although both offer marginal improvements against RoBERTa based embeddings. We are grateful for the reviewer for helping us to more carefully think about the robustness of our work in this regard (information on additional robustness checks now also features in the supplementary information).

Reviewer Three Minor Comment 17 (R3.25): *The authors claim on page 12 that SuperLearner “guarantees” that they can extract the “maximum predictive validity from the three data sources”. I don’t think SuperLearning makes any such guarantees in finite sample size (what they have here), but I do think it is a reasonable choice. I would suggest that the authors clarify this claim or remove it.*

Author’s Response: The referee is exactly correct to raise this slightly imprecise wording which we had previously used in the manuscript. Naturally, what we should have and do now claim is that SuperLearners create *optimal weights*, rather than guaranteeing the elimination of what others (i.e. Lundberg et. al 2024, PNAS) call ‘Learning Error’. SuperLearners do of course help in getting towards ‘guaranteeing’ that this learning error is eliminated, but do not guarantee maximum predictive accuracy from an estimated model. We have very much softened our language on this in accordance with our response above in Section 4.2.3 (‘SuperLearner’).

Reviewer Three Minor Comment 18 (R3.26): *Does the SuperLearner provide the weights for the weighted average that it uses? Could the authors report in the appendix, which models get most heavily weighted for which outcomes? These results seems like something that they already have and it might be useful in future research.*

Author’s Response: This is an excellent suggestion which hopefully much improves the paper. We have visualised these weights in a new Figure D.3, and comment on them in the appropriate section of our manuscript (Section 4.2.3); it is – perhaps surprisingly – the Linear Model and the Random Forest which are being allocated the highest weights amongst the six predictive algorithms which go into the Super Learner. We also allude to the visualisation of these weights in response to R3.20.

Comments From Reviewer Four

Reviewer Four Comment 1 (R4.1): *Moreover, the authors include related social commentary, albeit only briefly, by questioning the usefulness of genomics in future policy making. A subject not without controversy that needs the academic community to be more prominently featured in the debate in any context. Its inclusion is therefore appreciated.*

Author's Response: We are extremely grateful that the reviewer appreciated our inclusion of this perspective and commentary: it is an issue that we feel legitimately passionate about and agree that the academic community could be featuring more prominently in this debate.

Reviewer Four Comment 2 (R4.2): *Throughout the manuscript the authors use the word 'genes'. When referring to genotyped data, this term is incorrect since the information contained in the data goes down to single nucleotide polymorphisms (SNPs). Hence, it is necessary to refer to the genetic data of the NCDS sample as e.g., 'genetic markers' or 'SNPs'. when referring to polygenic scores, 'genetic predisposition' or 'genetic indices' for a certain phenotype (e.g., educational attainment, is more appropriate whenever neither their abbreviation ('PGS') nor their full name ('polygenic scores') are used. In titles, tables, and figures 'PGS' is the most appropriate and convenient term to use.*

Author's Response: We are grateful to the reviewer for pointing out our lack of precision. Where we had previously written 'genes' inline, we have replaced this with 'genetic markers', and within figures, tables, and section headings, we have replaced this with 'PGS' where appropriate.

Reviewer Four Comment 3 (R4.3): *The paper by Becker et al. (2021) in the footnote on page 12 needs to be properly cited and added to the 'References' section.*

Author's Response: We apologise for this oversight and corrected this. This was a typesetting error on our behalf.

Reviewer Four Comment 4 (R4.4): *The authors mention in the same footnote having obtained similar results when using LDpred-based PGS on preliminary analysis. Replication of results based on different methods strengthens the credibility of the paper's analyses and main conclusion regarding PGS. The manuscript would thus benefit from describing this in a bit more detail. The authors should consider expanding briefly on these additional analyses in supplementary sections.*

Author’s Response: We fully agree with the suggestion to provide more detail on the additional analyses using the LDpred-based PGS obtained from the PGI repo. In response to this insightful comment, we have expanded on these additional analyses to compare our results against those generated by PGS provided by the Polygenic Index (PGI) Repository (kindly see Supplementary Figure D.1). This repository provides LDpred-based PGS. These results – referenced in Footnote 1 and Supplementary Section A.3 – reflect this addition, where we also there emphasize the consistent nature of our results between the two types of PGS construction (and in fact, the LDpred based scores only improve prediction for one of the traits which we consider; Internalizing Behavior at age 16). Section A.3 also describes this process more fully.

Reviewer Four Comment 5 (R4.5): *The authors describe in section A.3 of the supplement the construction strategy of the PGS. While the fact that this data has been quality controlled and imputed is acknowledged, further details are not provided. Providing this additional information is advised since it would increase the work’s methodological transparency and would aid future replication attempts. The list of useful details entails but is not limited to the filters used to exclude low quality SNPs/samples and the imputation pipeline (if applicable, imputation servers used, well-established imputation workflows such as RICOPIII and settings therein such as filters according to imputation quality scores and MAF, version of the reference genome, etc). In the case that the above steps were not (entirely) carried out by the authors themselves since this is a dataset used in past research by many different groups, a reference that documented all details should be provided.*

and

Reviewer Four Comment 6 (R4.6): *The construction of the PGS used in the main analyses needs further clarification. The authors mention 33 different traits, yielding 33 different PGS but for the main analyses the reader is led to interpret that just one PGS feature was included. Was a composite measure utilized for the main analyses (e.g., a mean/sum score across all 33 PGS) or were all PGS entered independently and yet simultaneously to the SuperLearner analyses? If the latter is true, as the multiple-PGS approach cited in lines 139 to 140 would suggest, then why is just a single value reported on all main plots (and the range of variation indicated by the whiskers) and described in the results section? A short summary of the multi-PGS approach is needed around those same lines and/or in later sections.*

Author’s Response: Regarding the genotyping and imputation process, we have added more detailed information to Section A.3 of the supplement. As mentioned therein, comprehensive details about the quality control (QC) and imputation procedures can be found in the works of Artigas et al. (2015) and Davies et al. (2015). These studies provide information on the filters used to exclude low-quality SNPs/samples, the imputation pipeline, and other relevant parts of the process.

Regarding the construction and use of the PGS in our main analyses, we apologize for any confusion caused by our description. In our study, we utilized a multi-PGS approach, where all 33 PGS were entered independently and simultaneously into the SuperLearner analyses. Each PGS corresponds to a specific trait, such as cognitive performance, mental health, personality, and social behaviors. The SuperLearner algorithm is designed to handle multiple input features, including multiple PGS, and learns the optimal combination of these features to predict the target outcome. In our main analysis, we reported the overall predictive performance of the SuperLearner model, which takes into account the contributions of all 33 PGS. We made this clearer in the introduction section:

”We contrast and combine our text-based prediction with teacher assessments of students and state-of-the-art genomic prediction of multiple polygenic scores (see also Material and Methods).”

and the methods section:

”All polygenic scores were jointly used as input to the SuperLearner (see below), akin to a multi-polygenic score model (Krapohl, 2018).”

We have also naturally replaced PGS with its plural PGSs where appropriate.

Reviewer Four Comment 7 (R4.7): *The p-value threshold for SNP-inclusion in the PRSice algorithm was set to $p=0.5$. To set a single threshold might not be ideal given that the authors use summary statistics from a wide range of phenotypes due to the varying genetic architecture across traits. Consequently, the predictive power of these genetic measures is not well exploited. That is, the largest possible amount of variance of the outcome variables explained by the PGS is most likely not reached with the current settings. The authors could achieve more predictive genetic features by performing ‘best-fit PGS’ scoring already implemented in PRSice on each and every single summary statistics trait. The idea is to produce PGS at multiple thresholds, run linear regression with each PGS on the outcome variables and pick the p-value threshold at which the corresponding PGS explain the most variance. Finally, use the best-fit PGS to rerun the main analyses described in the main sections. Statistical overfitting represents no issue here since the authors apply nested cross-validation to control for it.*

Author’s Response: We sincerely appreciate the reviewer’s insightful suggestion regarding the optimization of polygenic scores (PGS) using the ‘best-fit PGS’ approach in PRSice. This is indeed a valuable method that could potentially enhance the predictive power of genetic measures. Our decision to use a single p-value threshold of 0.5 for all traits was based on maintaining consistency and comparability with other studies, as well as balancing computational efficiency with predictive power. While optimizing PGS construction for each trait could increase explained variance, it would also introduce additional complexity and computational demands that are beyond the current scope of our

study, which primarily focuses on comparing predictive performance across different data modalities. However, we fully agree that exploring different PGS construction strategies is a promising direction for future research. As such, we have added a note in our concluding paragraphs highlighting this as a valuable extension that could likely increase predictive accuracy:

“Furthermore, we have not data mined the three sources of information as thoroughly as we might have done, or utilised more emergent GPT models. An example of this is the fact that we created PGS using Clumping and Thresholding with a prespecified p-value threshold of 0.5 and we want to make our applications as standardised as possible; custom thresholds per trait would likely increase the predictive accuracy of our models even further. The same holds for more sophisticated Bayesian methods of PGS construction that leverage functional information about the genome (Zheng, 2024), which may further increase predictive performance, though not qualitatively change results.”

Reviewer Four Comment 8 (R4.8): *The issue of (genetic) population stratification n.11 eeds to be addressed since no mention of it is apparent in the manuscript. Associations between PGS and phenotypes are bound to be confounded by genetic similarity among individuals. The use of similarity measures can be included as covariates in the main analyses. Alternatively, before running main analyses, genetic similarity may be regressed out of the PGS via multiple regression. Genetic similarity can be operationalized as principal components extracted from the genotyped data using PLINK’s approach to principal component analysis or multidimensional scaling. The number of appropriate components to control for can be -determined e.g., by including only components that explain a vast proportion of genetic variance (as determined with a scree-test or with the Kaiser-Gutmann criterium). In the case that population stratification has been dealt with already, its procedure should be stated in the main text. On table E.1. depression is categorized as a personality trait. Depression is a mental disorder and should classified alongside ADHD, ASD, BP, and SCZ as a ‘Disease’.*

Author’s Response: We sincerely appreciate the reviewer’s thoughtful comment on population stratification, which is indeed an important consideration in genetic studies. We agree that this is a complex and nuanced issue in the field of genetics. Our approach in this study aligns with the perspective articulated by Plomin and von Stumm (2022) in their recent piece. They argue that from a predictive standpoint, population stratification can be viewed as a legitimate source of genetic variance contributing to polygenic score prediction within a defined population. This view suggests that once a phenotype and population are defined, any inherited DNA differences predictive of the phenotype in that population are valid sources for polygenic score prediction, regardless of whether they stem from ancestry, geography, or

culture. We acknowledge that population stratification is often treated as a confounder in genome-wide association studies and Mendelian randomization analyses aimed at inferring causality. However, our primary objective in this study was to assess the predictive power of polygenic scores in comparison to teacher assessments and textual data. Given this focus on prediction rather than causal inference, we chose not to correct for or remove genetic variance attributable to population stratification, as doing so could potentially reduce the predictive utility of the polygenic scores in our specific population. We appreciate the reviewer's suggestion to include similarity measures as covariates or to regress out genetic similarity from the PGS. While these are valuable approaches, particularly for studies focused on causal inference, we believe that they are beyond the scope of our current predictive analysis, and would conflate the message we are trying to directly deliver. However, we agree that explicitly stating our approach to population stratification in the main text would be beneficial for readers. We have accordingly added a brief explanation of our rationale in the methods section:

“Given our focus on predictive power rather than causal inference, we did not correct for population stratification, as doing so could potentially reduce the predictive utility of the polygenic scores in our specific population, following argument outlined by Plomin (2022), which considers population stratification as a valid source of genetic variance for polygenic score prediction within a defined population.”

References

- Artigas, S., M., L. V. Wain, S. Miller, A. K. Kheirallah, J. E. Huffman, I. Ntalla, N. Shrine, M. Obeidat, H. Trochet, W. L. McArdle, et al. (2015). Sixteen new lung function signals identified through 1000 genomes project reference panel imputation. *Nature communications* 6.
- Baeriswyl, F., C. Wandeler, and U. Trautwein (2011). Auf einer anderen schule oder bei einer anderen lehrkraft hätte es für's gymnasium gereicht: Eine untersuchung zur bedeutung von schulen und lehrkräften für die übertrittsempfehlung. *Zeitschrift für pädagogische Psychologie* 25 (1), 39–47.
- Davies, G., N. Armstrong, J. C. Bis, J. Bressler, V. Chouraki, S. Giddaluru, E. Hofer, C. A. Ibrahim-Verbaas, M. Kirin, J. Lahti, et al. (2015). Genetic contributions to variation in general cognitive function: a meta-analysis of genome-wide association studies in the charge consortium (n= 53 949). *Molecular psychiatry* 20 (2), 183–192.
- Dick, D. M. (2011). Gene-environment interaction in psychological traits and disorders. *Annual review of clinical psychology* 7, 383–409.
- Feigenbaum, J. and D. P. Gross (2020). Automation and the fate of young workers: Evidence from telephone operation in the early 20th century. Technical report, National Bureau of Economic Research.
- Filippova, A., C. Gilroy, R. Kashyap, A. Kirchner, A. C. Morgan, K. Polimis, A. Usmani, and T. Wang (2019). Humans in the loop: Incorporating expert and crowd-sourced knowledge for predictions using survey data. *Socius* 5, 2378023118820157.
- Foltz, P. W., D. Yan, and A. A. Rupp (2020). The past, present, and future of automated scoring. In *Handbook of Automated Scoring*, pp. 1–10. Chapman and Hall/CRC.
- Jones, S. E. (2013b). *Against Technology: From the Luddites to Neo-Luddism*. Routledge.
- Krapohl, E., H. Patel, S. Newhouse, C. J. Curtis, S. von Stumm, P. S. Dale, D. Zabaneh, G. Breen, P. F. O'Reilly, and R. Plomin (2018). Multi-polygenic score approach to trait prediction. *Molecular psychiatry* 23 (5), 1368–1374.
- Plomin, R. (2011, June). Commentary: Why are children in the same family so different? Nonshared environment three decades later. *International Journal of Epidemiology* 40 (3), 582–592.
- Plomin, R. (2018). *Blueprint: How DNA Makes Us Who We Are*. Mit Press.

- Plomin, R. and D. Daniels (1987). Why are children in the same family so different from one another? *Behavioral and brain Sciences* 10 (1), 1–16.
- Plomin, R. and S. Von Stumm (2022). Polygenic scores: prediction versus explanation. *Molecular psychiatry* 27 (1), 49–52.
- Power, C. and J. Elliott (2006). Cohort profile: 1958 British birth cohort (National Child Development Study). *International Journal of Epidemiology* 35 (1), 34–41.
- Risi, J., A. Sharma, R. Shah, M. Connelly, and D. J. Watts (2019). Predicting history. *Nature human behaviour* 3 (9), 906–912.
- Selwyn, N. (2021, December). *Education and Technology: Key Issues and Debates* (3rd edition ed.). London; New York: Bloomsbury Academic.
- Urhahne, D. and L. Wijnia (2021). A review on the accuracy of teacher judgments. *Educational Research Review* 32, 100374.
- van Loon, A. C. (2023). Predictability hypotheses: A metatheoretical and methodological introduction.
- Zellner, M., A. E. Abbas, D. V. Budescu, and A. Galstyan (2021). A survey of human judgement and quantitative forecasting methods. *Royal Society open science* 8 (2), 201187.
- Zheng, Z., S. Liu, J. Sidorenko, Y. Wang, T. Lin, L. Yengo, P. Turley, A. Ani, R. Wang, I. M. Nolte, et al. (2024). Leveraging functional genomic annotations and genome coverage to improve polygenic prediction of complex traits within and between ancestries. *Nature Genetics*, 1–11

We further delineate hereafter the changes which we have made to the appropriate parts of the manuscript in response to the all of the insightful comments of each reviewer in turn.

Comments From Reviewer One

Reviewer One Comment 1 (R1.1): *The authors have gone far above and beyond in addressing my original comments along with those from the other reviewers*

Author's Response: We are extremely grateful to the reviewer for their positivity here, and especially with regards to their appreciation of how much additional work went into our original resubmission.

Reviewer One Comment 2 (R1.2): *At this point in the process, I only have one additional comment regarding the methodology of this paper. Closed LLMs like the GPT series can be altered behind the scenes without our knowledge at any time, so I do worry about replicability. The authors partially address this by using multiple models, but in the final version I would implore the authors consider potential ramifications of their findings if, for any reason, the LLMs were to change in such a way that their results would not be replicable. At this stage I would not recommend any additional analyses, but the authors have a unique opportunity to sketch out these implications, and any insights would be valuable.*

Author's Response: We think this is a very reasonable point in the context of the fast-moving LLM landscape, and have addressed this insightful comment—albeit briefly—in two primary ways. Firstly, at the end of our data availability statement, we have appended the following: *‘Our embeddings are available upon request for the purposes of replication to accredited parties who have fulfilled all necessary compliance related steps’*. Secondly, we have added a new sentence into the concluding parts of our study which refers to the fact that these models are rapidly evolving, and situate this in the context of our GPT-3.5 vs GPT4 comparison: *‘Similarly – and while we found no discernible difference between the models which we employed – Large Language Models are currently evolving rapidly, and analysis done on closed models is subject to change at the behest of those who release their trained models’*.

Reviewer One Comment 3 (R1.3): *Beyond this I recommend this paper for publication.*

Author's Response: This is fantastic news. We again thank this and indeed all of the Reviewers; we believe the paper has improved immeasurably as a function of the process at Communications Psychology, for which we are grateful.

Comments From Reviewer Two

Reviewer Two Comment 1 (R2.1): *The paper has clearly been extensively and carefully revised in response to reviewers' detailed comments on the manuscript. The argument in the paper and the results are now much easier for a non specialist to follow and I believe that this will enhance the significance of the paper. I am not a specialist on NLP or genetic testing so I am not able to comment on these aspects of the paper but my in-depth knowledge of the 1958 cohort study gives me confidence that the authors have done an excellent job in leveraging the detailed longitudinal and multi-disciplinary data within the study. The detail in the paper is also likely to inspire further useful rigorous work that extends and expands the current analysis using different sets of variables, different outcome variables and increasingly sophisticated NLP.*

Author's Response: We are extremely grateful to the reviewer for their positivity, and believe we have no further action to take in this regard.

Comments From Reviewer Three

Reviewer Three Comment 1 (R3.1):

I would like to thank the authors for their improved manuscript. I believe that they sufficiently address the concerns of the reviewers, and I believe the paper should be published. I appreciated several of the improvements made by the authors such as adding more information about time (e.g., Fig 1), clarifying some analysis (e.g., Fig 3), and adding some new analysis (e.g., Fig D.3). I also liked the way they dealt with the fact that the GPT has and will continue to change over time; I found it interesting that there was little gain from GPT 3.5 to GPT 4. At this point I offer only suggestions/comments that I would fully leave to the authors. It is not my role as a reviewer to write the paper, and it will have the authors names on it (not my own).

Author's Response: We warmly receive the appreciation from the reviewer, especially with regards to their positivity in our actions to initially resubmit the manuscript. We are further grateful for the discretion that the Reviewer has given us regarding their suggestions for implementation. Please see our replies in general below.

Reviewer Three Comment 2 (R3.2): *I continue to think the title is misleading for the reasons described in my first review. I didn't find the authors' comments convincing, but this is their paper with their names on it.*

Author's Response: We are again grateful for concerns regarding the title, but are however going to exercise our discretion and chose to keep it as it was submitted. This is not to disregard the Reviewer's comment; we have discussed and carefully considered this at length as part of our processes.

Reviewer Three Comment 3 (R3.3): *I thank the authors for their improved attention to time in the text and figures, but I personally would still do more for the reasons described in my initial review.*

Author's Response: We appreciate this comment, and think that the issue is best resolved in the context of our first figure which introduces our research design. Specifically, we have appended an emphasis on the temporal dimension to the end of the sentence which introduces said figure: *'Our overall research design is schematically displayed in Figure 1, where we also additionally highlight the temporal element to our predictions.'* We have also improved all of the colourmaps in all figures, as well as specifically responding to a comment from Reviewer Four with regards to overlapping annotations.

Reviewer Three Comment 4 (R3.4): *I really like Fig 3, but the first time I read it I misunderstood it. The first time I read it I thought that the bars were cumulative in the sense that the second bar (simple indices of readability) includes that variable and previous variables (grammatical and typographical*

errors). I'm not sure why I had this misunderstanding or if it is worth trying to fix.

Author's Response: We have created a small addendum to Figure 3's caption: '*Note: the bars are not cumulative, but independent*'. We hope this avoids the potential confusion of any future consumers of our work.

Reviewer Three Comment 5 (R3.5): *My favorite part of the paper was section 2 (results). I felt like the section 1 (introduction) and section 3 (discussion) sometimes ranged far beyond what was in the paper and in a way that was not needed.*

Author's Response: While we agree entirely that the results of our work are the major contribution of it, we feel that the Introduction and (to a lesser extent) the discussion of it is necessary to both situate it within its context (e.g., the rise in automation), as well as in helping us to 'communicate more easily to the reader' our findings (see also **R3.6**).

Reviewer Three Comment 6 (R3.6):

The limitations section is important but a bit hard to parse now. There are many different things all packed into one paragraph. I wonder if they could group these into say a few categories and then signpost them more clearly (e.g., First,... Second, ... Finally...). I know the authors have thought a lot about the limitations, which I appreciate, and I wish their deep insights could be communicated more easily to the readers.

Author's Response: We thank the reviewer for this comment and adapted the limitation section accordingly.

Reviewer Three Comment 7 (R3.7): *I think it is interesting that Fig 3 seems to show that 1) we get more predictive accuracy from GPT 3.5 embeddings than any human extract features (e.g., index of readability) and 2) adding all human extracted features to the embeddings seems to add no predictability. I wonder if the authors think this is a general pattern or whether it might be specific to this dataset.*

Author's Response: While we do not think that this comment requires a clarification in the manuscript, we would state for the record that we believe this pattern is likely to generalize beyond our specific dataset. In fact, given that our analysis uses relatively short text samples, we would expect LLMs to demonstrate even stronger performance advantages over traditional metrics when analyzing longer texts, where their ability to leverage contextual knowledge and understand complex linguistic patterns can be more fully expressed. The lack of additional predictive power when combining embeddings with human-extracted features suggests that LLM embeddings are already capturing these simpler metrics while also encoding richer semantic and structural information. This observation raises an interesting methodological question for future research: rather

than using embeddings as input features for downstream prediction tasks, it may be more effective to leverage LLMs' capabilities directly through careful prompting or fine-tuning to generate predictions from text. This would allow more direct use of their sophisticated language understanding capabilities, though such an approach would require careful validation and comparison with more traditional methods like the embedding + superlearner approach used in our study.

Reviewer Three Comment 8 (R3.8):

Again, I offer these pieces of feedback with the hope that they help the authors. Ultimately, I would leave it to them to decide. Overall, I think this is an interesting paper, and that it was improved by this round of revisions.

Author's Response: We are again grateful for the discession that Reviewer Three has afforded us.

Comments From Reviewer Four

Reviewer Four Comment 1 (R4.1): *I thank the authors for addressing the raised concerns pertaining the previous manuscript submission with such care. For the most part, adequate changes were introduced to the revised version, either by supplying additional information or by providing reasonable arguments to support analytical choices. However, there is one major point in need of more thorough examination, alongside a few very minor ones, the latter being concerned merely with formalities in the way information is reported/displayed. As such, the following is divided into “major” und “minor” comments.*

Author’s Response: We thank the Reviewer for their thoughtful, diligent review at this second round, and believe that we are able to provide answers to the ‘major’ and ‘minor’ comments below accordingly.

Reviewer Four Comment 2 (R4.2): *In the revised manuscript, as well as in the response to the previous comments, the authors take the informed decision to abstain from assessing and mitigating the effect of population stratification in analyses involving PGSs. The standpoint to support this decision proposes that genetic variation stemming from genetic ancestry is a legitimate source of PGS prediction. While this sounds reasonable since PGS are constructed based on the very fact that people vary in their allele frequencies in all polymorphic loci (which is determined to a substantial degree by ancestry), information contain in genetic load on traits are dissociable from that contained in genetic ancestry. Between closely related individuals (e.g., siblings) inherited genotypes will not be identical since they arise due to random independent meiotic events, which are not influenced by population stratification¹. It is this portion of signal, as opposed to that shaped by genetic ancestry, that is the subject of academic research that tries to optimize genetic scores for prediction and possibly for future disease risk assessment in health care settings. Leaving population stratification uncontrolled for exacerbates the already existing problem of interpreting what PGS truly measure. Work by David Curtis² in 2018 serves as an example. The study concluded that both magnitude and distribution of Schizophrenia PGS were highly dependent on ancestry. Individuals of African descent had a 10-fold higher genetic risk for schizophrenia (as measured with PGSs) than European individuals, when the PGS were constructed using GWAS summary statistics derived from European samples. In all likeliness, this does not mean that people of African genetic background are at higher risk, since the difference in PGSs between individuals with- and without a schizophrenia diagnosis was still significantly different from 0 after controlling for population stratification and since the disease prevalence is roughly the same across populations worldwide. Furthermore, the PGS relationship with ancestry has been repeatedly found to reduce greatly the already weak predictive performance of PRS, when GWAS summary statistics*

are applied for PGS construction to genotyped data from individuals that do not match the genetic ancestry of the discovery GWAS³. The PGSs' performance varies even between individuals of the same ancestry⁴, though to a smaller degree. Thus, biased performance estimates due to this kind of confounding is very likely without stratification correction/control in the present manuscript since the GWAS summary statistics used were obtained from studies with populations of varying ancestry. It should be noted, however, that since the population under investigation in the present manuscript is exclusively British, the impact of confounding due to ancestry is likely to be less than in multi-ancestry or non-european studies. Nevertheless, not negligible. As the authors point out in their response to my comments, confound is of interest in studies aiming at causal inference, but as laid out above, this applies for studies focusing on prediction as well. I therefore request the report of sensitivity analyses that compare the results in the presence and absence of statistical control of genetic ancestry. As mentioned in the last revision round, principal components derived from PLINK's "--pca" command offer a good solution, but the choice of technique to operationalize this construct is of course left to the authors. I recommend including the first few principal components (e.g., four, or an empirically supported number. For the latter, see recommendations on the last revision round) as covariates in PGS models, but reporting the fraction of total explained variance that is solely attributable to the PGS (if programmatically feasible, otherwise, kindly refer to the last round of revisions for other alternatives). Finally, the authors express their concern for performance drops after ancestry control. Regardless of potential performance drops, the manuscript's scientific value could benefit from this procedure because the interpretability of the PGSs and, by extension, of the results, would become easier and alternative explanations questioning the results' validity would be ruled out.

Author's Response: We are grateful to the reviewer for their thorough comments regarding population stratification. Our analysis focuses exclusively on the 1958 British Birth Cohort, a demographically homogeneous sample from an era of British history with well-documented population characteristics. While the reviewer's discussion of cross-ancestry prediction raises important considerations – including the cited schizophrenia example (though we note that recent meta-analytic evidence actually supports substantially elevated risk of schizophrenia among African-Americans, with van der Ven et al. (2024) reporting OR = 2.07 [95% CI: 1.64-2.61]) – these methodological concerns are not directly applicable to our historically specific sample. We further again emphasize that our study's objective is prediction, not causal inference. While the methodological considerations from Mendelian randomization studies are valuable in their appropriate context, their application to purely predictive analyses requires careful consideration of the different underlying aims. Still, to address these methodological questions comprehensively, we conducted two additional sensitivity analyses, which show the absolute lack of any predictive power when including ten principal components. First, we estimate in-sample

models with simple linear regression on each trait and calculate the R^2 metric as follows:

Outcome	R-squared	Adj. R-squared	P-value	Nobs
General Factor of Cognitive Ability (Age 11)	0.00241	0.000628	0.196	5614
Verbal Ability (Age 11)	0.00173	-0.000522	0.467	5617
Nonverbal Ability (Age 11)	0.00310	-0.000132	0.0658	5617
Reading Ability (Age 11)	0.00144	-0.000344	0.622	5615
Mathematical Ability (Age 11)	0.00311	0.00133	0.0650	5615
Reading Ability (Age 16)	0.00274	-0.000996	0.190	5007
Mathematical Ability (Age 16)	0.00272	-0.000719	0.195	4985
Occupational Aspirations (Age 11)	0.00422	0.00227	0.0133	4563
Scholastic Motivation (Age 16)	0.00191	-0.00135	0.500	4937
Externalizing Behavior (Age 16)	0.000974	-0.00104	0.489	4873
Internalizing Behavior (Age 16)	0.00183	-0.00158	0.513	5039
Highest Education (Age 33)	0.00241	0.000631	0.195	5619

We next re-ran our SuperLearner for each trait with ten Principal Components, seeing an almost identical lack of explanatory power:

Outcome	Mean R-squared (Min, Max)
General Factor of Cognitive Ability (Age 11)	-0.0002 (-0.0071, 0.0053)
Verbal Ability (Age 11)	-0.0001 (-0.0025, 0.0028)
Nonverbal Ability (Age 11)	-0.0002 (-0.0022, 0.0007)
Reading Ability (Age 11)	-0.0001 (-0.0014, 0.0017)
Mathematical Ability (Age 11)	-0.0017 (-0.0216, 0.0064)
Reading Ability (Age 16)	0.0018 (-0.0017, 0.0074)
Mathematical Ability (Age 16)	-0.0013 (-0.0045, 0.0008)
Occupational Aspirations (Age 11)	-0.0013 (-0.0124, 0.0012)
Scholastic Motivation (Age 16)	-0.0018 (-0.0070, 0.0017)
Externalizing Behavior (Age 16)	-0.0659 (-0.1197, -0.0017)
Internalizing Behavior (Age 16)	-0.0178 (-0.0731, 0.0069)
Highest Education (Age 33)	-0.0013 (-0.0132, 0.0030)

These results demonstrate that genetic PCs alone achieve consistent near-zero predictive power across all outcomes, providing clear empirical evidence regarding the role of ancestral stratification in this context. As a sign of respect for the thoughtful and considered suggestion of the Reviewer, we now include these two tables in the appropriate part of the Supplementary Information (Supplementary Tables E2 and E3), and reference them accordingly in the text (i.e., footnote 2 in Section 2.3).

Reviewer Four Comment 3 (R4.3): *In the footnote on page 15, while the typesetting error was removed, the paper itself is still not cited correctly. It should read “[...] the extensive set of polygenic scores provided by the Polygenic*

Index Repository (Becker et al., 2021), constructed using LDPred (Privé et al., 2020), was also tested in preliminary analyses [...]”

Author’s Response: We thank the reviewer for their sharp look at our references and fixed them for the respective footnote.

Reviewer Four Comment 4 (R4.4): *According to the footnote on page 15, the method is ‘LDPred2’ but on line 915, ‘LDPred’ is mentioned. Please check which of both methods was used.*

Author’s Response: We thank the author for this observation, it is indeed LDPred and we corrected it.

Reviewer Four Comment 5 (R4.5): *Figure D.1 of the supplements: This addition is much appreciated. Figure readability could be improved by making sure that the text of the holdout mean R^2 annotations don't overlap*

Author’s Response: We have improved the colour maps of all of our figures without changing any of their content. This builds upon suggestions given by previous rounds of reviews, such as why colormaps were not a singular colour when there were, for example, not multiple classes of target variables being plotted at once. We have also explicitly fixed this annotation issue in D1 by improving the layout positioning of the text.

Reviewer Four Comment 6 (R4.6): *Table E.1 of the supplements: As per my comment during the first round of revisions, depression should fall under the category “disease” rather than “personality trait”.*

Author’s Response: We thank the reviewer for this comment and adapted the table accordingly.

I further delineate hereafter the changes which I have made to the appropriate parts of the manuscript in response to the all of the last comment of reviewer 4.

Comments From Reviewer Four

Reviewer Four Comment 1 (R4.1):

I want to highlight the author's attention to the smallest of details and issues, a testament of their professionalism. Their dedication devoted to my past comments is no exception. The effort put by the authors to address the effect of population stratification in the sensitivity analyses is much appreciated. I think conclusions drawn from them provide valuable additional information that increases the validity of the results reported in the main text. As the authors point out in their answers to my past comments, these analyses show indeed a lack of predictive power of the genetic principal components alone and imply an inconsequential impact of population stratification on the PGSs' predictive capabilities in the sample at hand. However, the principal components are meant to be handled as covariates to the rest of predictors of interest (i.e., the PGS), and not as the only regressors present in the predictive models since their unique contribution to prediction performance is of less interest. According to footnote 2 of section 2.3, the sensitivity analyses were carried out doing the latter. These covariates are to be included alongside the PGS in the statistical models of the sensitivity analyses. The performance reported should be that of the PGS within those models or, if not programmatically possible, of the whole model. The supplementary analyses reported in Tables E2 and E3 should be updated to reflect these changes. Additionally, for the sake of clarity, the title of supplementary table E3 should explicitly refer to the usage of PGSs and ten principal components in the SuperLearner models. This way, the predictive performance of the PGSs can be clearly contrasted between models with- (in a revised version the supplementary tables) and without (as already reported in the main text without any need of further changes) population stratification control. After this update, I firmly believe that the manuscript will be suitable for publication.

Author's Response: We are sincerely grateful to the reviewer for their meticulous attention to detail and highly professional and constructive feedback throughout this process; their dedication is much appreciated. We particularly value the detailed comments regarding the handling of potential population stratification.

Following the reviewer's explicit guidance, we have now performed the requested sensitivity analysis by including the first ten genetic principal components as covariates alongside the full set of polygenic scores within the SuperLearner framework. The results of this analysis, assessing the predictive performance of the PGSs while controlling for population stratification in this manner, are now presented in the newly added Supplementary Table 4.

As anticipated for this historically specific and relatively homogeneous sample, the inclusion of these principal components alongside the PGSs resulted in negligible changes to the

predictive performance reported in the main text, confirming the robustness of our findings. We thank the reviewer for encouraging this valuable addition, which further strengthens the validity of our results, and trust this addresses their final concern.